# The secret life of baby turtles: A novel system to predict hatchling emergence, detect infertile nests, and remotely monitor sea turtle nest events

Erin B. D. Clabough[1]*, Eric Kaplan[2], David Hermeyer[3], Thomas Zimmerman[4], Joshua Chamberlin[5]¤, Samuel Wantman[3]*

1 Department of Psychology, University of Virginia, Charlottesville, VA, United States of America, 2 Hatteras Island Ocean Center, Hatteras, NC, United States of America, 3 Nerds Without Borders, San Francisco, CA, United States of America, 4 IBM Research-Almaden, San Jose, CA, United States of America, 5 Department of Biology, Hampden-Sydney College, Farmville, VA, United States of America

¤ Current address: College of Veterinary Medicine, North Carolina State University, Raleigh, NC, United States of America

* ebd2r@virginia.edu (EBDC); sam@wantman.net (SW)

**Data Availability Statement:** All the data underlying the results found in this paper can be found on the Nerds without Borders website: http://

## Abstract

Current understanding of sea turtle nesting, hatching, and emergence events has been largely limited to observable events on the surface of the sand, though recent approaches using audio or visual equipment have allowed scientists to better understand some underground nest phenomena. We used a technology-based approach to define motion-related *Caretta caretta* hatching and emergence nest events. We describe a novel low-cost, accelerometer-based system called TurtleSense that can detect movement and temperature within sea turtle nests remotely. TurtleSense is successfully able to specifically detect motion within sea turtle nests over the entire course of incubation. This system allows for the identification of infertile nests and the detection of four predictable sequential developmental activity patterns in viable nests, including a hatch and posthatch period, the timing of which can be used to tightly predict hatchling emergence events almost to the day. TurtleSense provides a much better understanding about what is happening in the nest before emergence and allows for the generation of a theory of the mechanism that triggers mass emergence. Our results suggest that motion plays a large role in hatchling communication and that the timing of emergence events may be related to the cessation of movement within the nest. Current management of sea turtle nesting events is primarily driven by counting the number of days since the nest was laid, with further safeguards placed at the nest upon subsequent visual observation of depression or emergence events. Use of TurtleSense technology can impact nest management and conservation efforts, allowing organizations to use this motion data to more tightly predict emergence dates for sea turtle hatchlings and to use viability data to inform nest management decisions.

nerdswithoutborders.net/index.php?title=Turtle_Sense_raw_data Each year of data is compressed into a single zip file. The instructions for creating the devices and the code are on the Nerds without Borders website: http://nerdswithoutborders.net/index.php?title=Turtle_Sense/Phase_Two.

**Funding:** This work was supported by volunteer work, donations, and by agreements listed in the acknowledgments section between the Hatteras Island Ocean Center nonprofit and the National Park Service. This work was also supported by the Virginia Foundation for Independent Colleges in the form of a grant awarded to EC. This works was also supported by the Hampden-Sydney Honors Program in the form of a grant awarded to JC.

**Competing interests:** The authors have declared that no competing interests exist.

## Introduction

Green, Hawksbill, Kemp's Ridley, Leatherback, Loggerhead, and Olive Ridley sea turtles are threatened species, and as such, their nests are protected by law in the U.S. Endangered Species Act in the United States [1]. Sea turtle populations worldwide are in decline because of assorted threats, most of which are attributed to anthropogenic pressure, including development and loss of nesting habitat, fishing, poaching, climate change, and ocean pollution [2, 3].

Newly hatched sea turtles (hatchlings) are also highly susceptible to predation once they leave the nest [4]. It has long been known that sea turtles will move toward artificial lighting instead of orienting to the sea, resulting in misorientation near roads and human housing, a problem that has only intensified as human activity increases [5, 6]. To this end, the National Park Service (NPS), local organizations, volunteers, and non-profit organizations cooperate to generate unique protocols for how sea turtle hatchlings are best protected on various beaches.

Conservation efforts largely concentrate on protecting the more vulnerable juvenile population rather than the existing mature population [7, 8]. Nest monitoring and protection constitutes a large portion of sea turtle nest management resources, as the hatchlings face a myriad of dangers on a nesting beach [9]. Accurately knowing when the hatchlings will make their initial journey from the nest to the sea is key to protecting hatchlings from predators and human beach traffic, as well as preventing misorientation to artificial lighting. In order to better predict emergence dates, the factors leading to hatchling emergence need to be better understood.

It has been known for some time that sea turtle hatchlings work together to dig their way out of the nest [10]. There are multiple advantages to emerging en masse, including 1) speeding up escape from the nest, 2) predator dilution since the more turtles come out at once, the more protection it offers from predators, and 3) the more turtles present, the less energy is used because they share the digging work [11]. Several factors contribute to the timing of hatchling emergence from beneath the sand, specifically temperature, sand conditions, nest depth, and number of hatchlings. Most notably, high surface sand temperatures during the day inhibit hatchling digging activity toward the surface, impacting the timing of emergence [12–15]. Hatchlings emerge earlier in the night from shallow nests than from deeper nests, which is correlated to more rapidly cooling sand [16]. Secondly, the time between hatching and emergence can vary because of the depth of the nest and the compaction of the sand [17]. The general consensus in the field is that the further the turtles have to dig, the longer emergence takes, and the fewer turtles in the nest, the slower they will emerge, but motion monitoring has not taken place in a large scale way in nests to confirm that digging actually takes more time [11].

Most sea turtle hatchling studies focus on isolating emergence cues or orientation cues towards the sea after emergence, yet synchronous emergence is not the same thing as synchronous hatching and it has remained unclear whether the presence of a mass coordinated emergence event is due to delayed emergence rather than synchronized hatching [18]. The hatching process has been thought to begin approximately 4 days prior to the emergence event in loggerhead turtles [*Caretta caretta*] and presumably in other species of marine turtles as well [19].

Scientists have often wondered how hatchlings coordinate their activity in order to leave the nest together. Evidence points to environmentally cued hatching, perhaps based on more ideal temperature and oxygen conditions (reviewed in [20]). However, this type of social facilitation could also be facilitated by auditory information received by the hatchlings. Green sea turtles (*Chelonia mydas)* perceive vibrations using unique, fibrous stapedosaccular strands hypothesized to relay vibrational energy [21] and we know that *C. caretta* turtles respond to low-frequency sounds and vibrations [22].

Although vocalization has recently been postulated as a mechanism underlying synchronous hatching in some birds and reptiles, it seems less likely to be used in a relatively non-

vocalizing species like the sea turtle. Kemp's Ridley (*Lepidochelys kempii*) turtle hatchlings can vocalize in the nest [23]; however, no significant differences are seen in vocalizations between incubation, hatching, and emerging from the nest [24], making it unlikely that the vocalizations serve as a coordinator of hatchling activity.

Temperature variation, including egg location within the nest, can alter individual incubation times, but synchronized hatching activity does not appear to ebb and flow with temperature [18, 25]. There is no conclusive evidence for synchrony cues besides temperature, but synchronized hatching involving active embryo-to-embryo communication can occur in freshwater turtles [18]. It is thought that sea turtle hatchlings delay emerging from the nest for more than two days after hatching from their eggs [17, 26], but little is known about hatchling activity during this crucial posthatch period right before emergence.

It has long been maintained that sea turtle hatchlings can wait in an inactive state just below the nest surface until the ideal emergence time, sometimes even with their heads above the sand [13, 27]. It appears that the stillness of the uppermost turtles has a dampening effect on the activity of hatchlings below until a temperature drop indicates night has come and it is safer to head towards the sea. Even during the day, experimental removal of the surface turtles leads to deeper nest activity and emergence within minutes [13]. Thus, motion within the nest from sea turtles moving within the eggs and hatching can potentially stimulate nestmates to hatch and/or emerge, but motion data have not been extensively analyzed in sea turtles.

Accurately predicting hatching emergence is important because the incubation window can vary greatly depending on the time of year and geographical location. In Zakynthos, Greece, incubation times on different beaches averaged from 51–70 days between 1984–2002 [28]. On Cape Hatteras in North Carolina in 2018, sea turtle hatchlings typically emerged between 52–64 days after the nest is laid [29]. This multiweek potential emergence window is a tedious time of uncertainty because the current method for monitoring sea turtle hatchling emergence typically consists only of two elements: 1) Timing the number of days since the nest was laid and 2) Manually checking for a sand depression above the nest, which is a sometime indicator that eggs are in the process of hatching. During this window in some locations, volunteers often sit with vulnerable nests each evening to ensure hatchlings are not disoriented by light pollution upon emergence. In addition, expanded protective barriers are installed around turtle nests, and beach closures can occur, negatively impacting beach goers, tourism, and the fishing industry.

Hatteras sea turtle nesting activity has not been well described previously in the literature, despite the fact that the continental shelf near Cape Hatteras is a loggerhead sea turtle foraging hotspot and Hatteras supports a growing number of yearly sea turtle nests [30, 31]. Cape Hatteras National Seashore is home to large tourist and fishing industries that depend on beach use. As the popularity of Hatteras Island National Seashore has grown, hatchlings that emerge on Hatteras Island can be misoriented by campfires and artificial lighting on human structures, especially near the villages. In addition, the presence of off road vehicles may affect the beach profile and substrate characteristics in a way that reduces suitability for nesting and reduces emergence success [31].

In recent years, conflicts have emerged in the Seashore over necessary beach closures due to conservation-related nest protections, as the system of expanding nest enclosures based on nest age in order to keep hatchlings safe forces closures of sections of the beach and therefore limits recreational use of the Seashore and impacts the livelihood of fishermen who depend on beach access. Shortening the length of time these barriers are required would have enormous community benefit, but to meet this goal, more information about hatchling emergence dates is necessary.

A National Research Council report concluded that advances in turtle population ecology will come primarily from improvements in monitoring systems [32]. To better understand nest motion patterns during incubation, we developed nest sensors that transmit accelerometer data wirelessly through cell phone towers in order to monitor nest events from afar, with the goal of tightly pinpointing emergence dates and enabling the erection of "just-in-time" barriers. Our sensor and data acquisition system, called TurtleSense, is capable of detecting turtle nest hatching activity. Previous field testing has streamlined sensor selection, wireless networking, low-power design, data processing, and environmental packaging for efficient use in the harsh environment of a beach [33].

The current study describes the use of the TurtleSense remote monitoring system over several nesting seasons on Cape Hatteras National Seashore. The goals of our study were to better understand and define motion-related sea turtle hatching and emergence nest events, examine the impact of nest depth and hatchling number on emergence dates, and to use collected motion data to more tightly predict emergence dates for sea turtle hatchlings.

## Methods and materials

### TurtleSense hardware

The TurtleSense system consists of two main components: 1] a sensor that is placed within the nest, and 2] an external communication tower located adjacent to the nest. The sensor circuitry includes an accelerometer, temperature sensor, and microprocessor on a 25mm X 25mm circuit board (as fully described in [33]). The accelerometer records any changes in motion detected within or around the nest, and the temperature sensor records the temperature fluctuations within the nest (˚C). The accelerometer can be remotely programmed to record and process data between 12.5–400 times/sec. The smallest recordable temperature fluctuation is 0.1˚C. The circuit board is sealed inside a polyurethane sphere designed to keep moisture out and resemble the size, shape, and coloration of a sea turtle egg (sensor egg). The sensor egg is attached to a 6-meter cable that ends with a connector that plugs into a communication unit. The communication unit contains a microprocessor, batteries and a cell phone module sealed in a PVC pipe tower (76mm in diameter) to protect it from any harsh weather conditions. In addition, each nest is initially registered using a handheld device, which contains a portable duplicate of the communications unit circuitry [34]. The plans for creating the circuitry and assembling the units are available online [35]. The cost of parts is approximately $300 per unit to build independently ($50/sensor and $250/communication tower).

### TurtleSense software

During the initial nest registration, the handheld device is programmed to test the sensor, and record the date and GPS location into the microprocessor in the sensor. This process is done when the sensor is installed and does not require the presence of the communication tower. The date when the sensor is registered and the serial number of the sensor is used later to identify the specific nest in every report.

The communication tower is typically installed between day 35–40. Initial tests found that the readings for the first 40 days were virtually the same as background noise, so delaying communication tower installation allows the communication devices to be used for shorter periods of time and thereby service more nests. Once connected to the buried nest sensor, the communication tower can begin to transmit data. While the sensor is set to record motion 100 time a second, the raw data are summarized regularly at intervals that can be set from every 15 seconds to 6 minutes. The sensor sends data to the communications unit when ready to upload a

report to the internet. This occurs when there are 240 records, which is either one day (using 6 minute records) or 4 hours (using 1 minute records).

When the sensor alerts the communication unit that it is ready to send data, the microprocessor within the communication unit turns on the power supply for the cell phone board and turns on the transceiver chips that send the data over the Cat5e cable. A text file report is uploaded to a server on the internet, readable by humans and computers. Once the data are uploaded to the online server, the transceivers are turned off and the sensor goes back to collecting data.

## Processing of change in motion data

Each accelerometer reading is a vector that measures the magnitude and direction of the force acting on the sensor at the time of the reading. The readings are reported as X, Y and Z Cartesian coordinates, and each coordinate can vary from positive 2 g to negative 2 g in magnitude. Each coordinate is reported as a 12-bit signed number, so the lowest bit corresponds roughly to a force of 0.001 g. If the sensor is static (not moving), the magnitude of the X, Y and Z readings of the accelerometer roughly correspond to 1 g of force and should have a magnitude of about 1024.

The sensor constantly analyzes the readings from the accelerometer to create a profile of the energy from the motion in the nest over time. The microprocessor compares successive readings from the accelerometer to calculate the change in acceleration [or "jolt"]. The jolt is calculated by computing the magnitude of the difference of the two acceleration vectors. The formula for the calculation is: $Jolt_n = \sqrt{(x_n - x_{n-1})^2 + (y_n - y_{n-1})^2 + (z_n - z_{n-1})^2}$.

The microprocessor can easily and quickly add and subtract and has a built-in math coprocessor, so except for the square root function, the *Jolt* can be easily and quickly calculated. Because calculating a square root is quite complicated, we do not store information about the magnitude of the jolts recorded. Instead, a calculation for the square of the *Jolt* ($Jolt^2$) is done: $(Jolt_n)^2 = (x_n - x_{n-1})^2 + (y_n - y_{n-1})^2 + (z_n - z_{n-1})^2$.

The magnitude of each $Jolt^2$ is placed on a logarithmic scale divided into 29 different ranges (based on the highest bit of the 29 bit binary number from the $Jolt^2$ calculation). The program has counters for each of the 29 different ranges. Each counter can be thought of as a bin. If a magnitude is in range, it is added into the bin for that range. After the programmed amount of time (typically a few minutes), the bin counts are stored in a record along with a temperature and orientation reading, and a new set of bin counters is started. Each record provides a profile of what happened during that period of time.

The number of readings in each bin reflect the jolt information that occurred while the bins for that record were being filled. The reports uploaded to the TurtleSense server are 20,834 bytes/day/nest, resulting in a compression ratio of approximately 2500:1. However, even when set at 6 minute intervals, the 240 records created each day provide adequate information about daily nest events.

In order to easily read the motion information, these approximate average energy levels ($Jolt^2$) over a preprogrammed period of 1–6 minutes are consolidated into graphical data for each nest over time. Due to the small scale of the readings, only very small motion changes are graphed. Large disturbances, like from predation or human disturbances, are way off the chart. These graphs are maintained online and updated in real time as the reports come in (A data flowchart is depicted in Fig 1). In effect, the data processing integrates the raw data, providing a profile of the total energy of the motion that is sensed.

## TurtleSense field testing

This work was approved by the North Carolina Wildlife Resources Commission Division of Wildlife Management (Endangered Species Permit 14ST74) and performed in cooperation

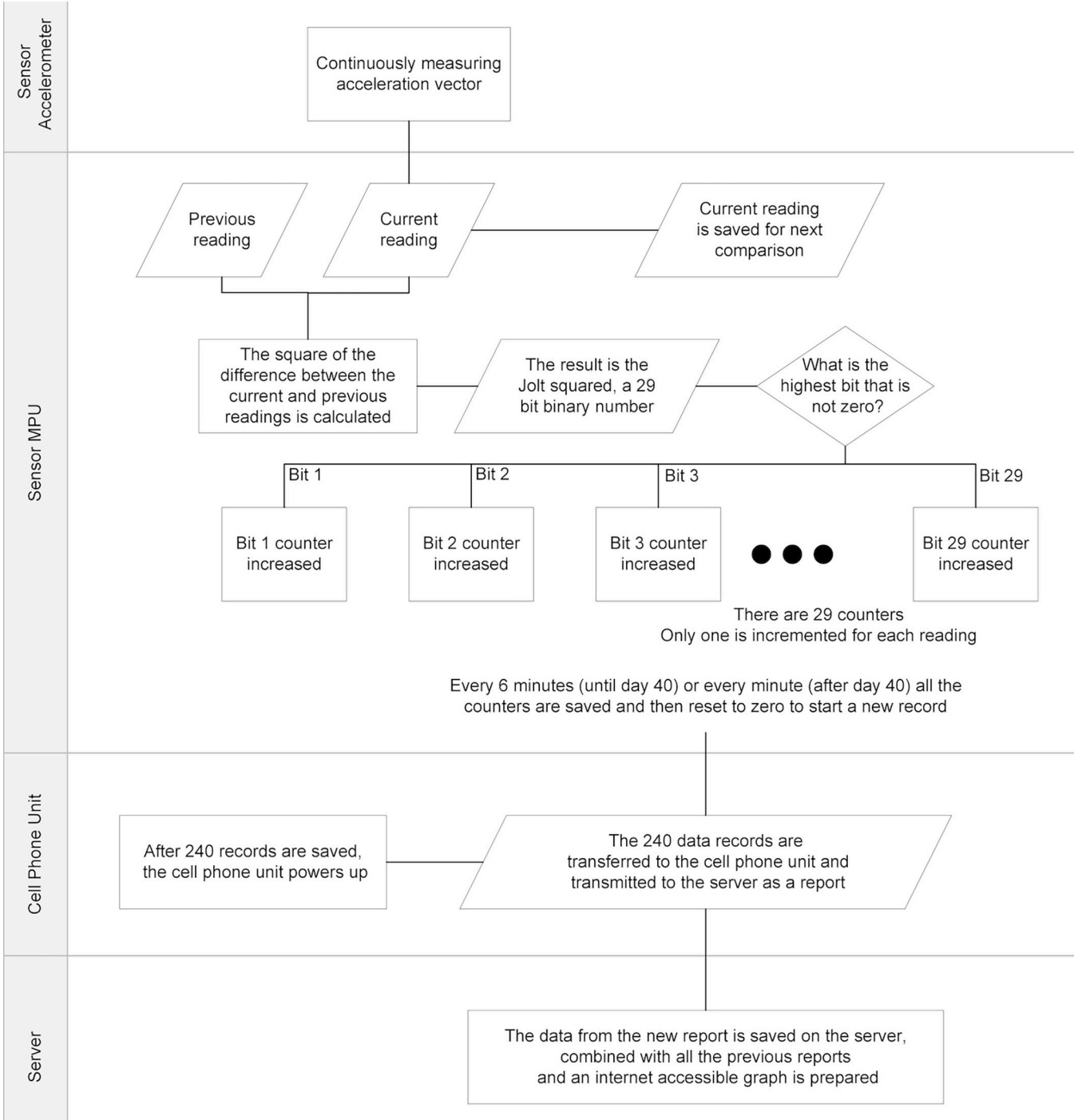

**Fig 1. Schematic of data processing.** A motion detector in the sensor placed in the nest constantly measures and computes changes in acceleration. The Micro Processor Unit (MPU) in the sensor counts the readings in 29 counters after sorting them by magnitude based on powers of 2. After a period of either one or six minutes, the 29 counts are stored as a record and the counters are reset. After 240 records have been created, the Cell Phone Unit is turned on and the data are sent to a dedicated server. The new data are stored, combined with the previous data, and graphed for internet access.

with the US Department of the Interior National Park Service (Cape Hatteras) under Park Assigned Scientific Research and Collecting Permit numbers CAHA-2013-SCI-0016 (Zimmerman)—CAHA-2017-SCI-0013 (Clabough). Normal turtle nest management protocols outlined by the National Park Service were followed, with the addition of our sensor installation protocol.

During 2013–2017 nesting seasons, 74 sea turtle nests were monitored with TurtleSense equipment on the Cape Hatteras National Seashore located on the Outer Banks of North Carolina (USA). The same basic placement protocol was used each nesting season, but modifications were made to the hardware each year to adapt the system to the harsh environmental conditions and increase the ease of data collection.

In 2013, a few nests were monitored using a "proof-of-concept" prototype to test the basic viability of the project, though no usable data were collected during this period. Following this initial field testing, we designed the hardware and software systems used to collect the data. Over several nesting seasons, we improved data transfer over longer cables, lowered the power drain on the system, and optimized equipment to better withstand over-wash, storm, moisture, sand, and shipping conditions. This development work continued, resulting in current units that are solar-powered or equipped with rechargeable batteries and require less maintenance, with field repairable features.

## Sensor and communication unit installation

Smart sensors were placed within *Caretta caretta* sea turtle nests (both *in situ* and relocated nests). The sensor identification number, nest identification number, and nest location were recorded before the sensor was installed at each site. When a newly laid nest was located, the smart sensor GPS location and cell reception were registered using a handheld communications device. The top ten eggs were carefully removed from the nest. The smart sensor cable was coiled once around the inner wall of the nest and the sensor was placed in the center of the remaining eggs. This coiled placement near the top of the nest allows the sensor to move down with the eggs as they hatch. The ten eggs were replaced around the sensor and the nest was covered with surrounding sand.

The sensor cable was buried in a sand trench at the same depth as the nest heading towards the back right dune corner of the nest enclosure, where the remaining cable was coiled around a wooden stake until communication tower placement. Nesting sites were checked daily throughout the incubation period to document hatching events, including the presence of depressions, and/or nest disturbances. Other than daily visual inspection, nests were not monitored until communication towers were placed.

Communication towers were installed anchored in buckets filled with concrete buried in the sand at the back right dune corner between day 35–40 (or when available) and connected to the smart sensors. A few initial nests were monitored from Day 1, while other nests were monitored only later in the season (in order to maximize the number of nests being monitored). After the communication tower was placed, the TurtleSense graph for each monitored nest was observed daily for initiation of hatching activity.

## Hatchling emergence

Emergence numbers were quantified by visual observation if nest sitters were present in the evenings and/or based on number of hatchling tracks during Turtle Patrol nest visits each morning. Per National Park Service (NPS) protocol, nests were excavated 3–5 days after an initial emergence event. Any live hatchlings discovered upon excavation were collected and released the same day at or after dusk. All communication units and smart sensors were

removed during excavation. Actual emergence data are based on real-time human observations and morning NPS hatchling track counts.

## Using TurtleSense to detect activity and predict emergence events

Using TurtleSense, motion readings from 79 nests were analyzed over a period of 5 years (2014–2018; 72 *Caretta caretta*, *2 Chelonia mydas, and 5 Lepidochelys olivacea)*. From these readings, motion data were analyzed for this study in 72 monitored *Caretta caretta* nests (both relocated and *in situ*) on Hatteras Island, NC in 2014 (n = 20), 2015 (n = 29), 2016 (n = 12), and 2017 (n = 11). Nest depth (to the top of the clutch) was recorded when the sensor was placed for *in situ* nests. Of the nests used in this study, 9 were relocated due to original placement location below the average high tide line where regular inundation would have resulted in embryonic mortality. These nests were reburied as close as possible in areas above the high tide line relatively free of vegetation at the same nest depth/dimensions, and with the same egg layer placement.

Our accelerometer took readings at 100 times/sec. Reports for our collected data were summarized every six minutes if connected before Day 40 and then every minute from Day 40 on. Graphs were automatically generated with sensor data once a day for the first 40 days and then every four hours after that so that nest data were analyzed for patterns of activity throughout development. Two *Chelonia mydas* nests were also monitored on Hatteras Island to confirm activity patterns, though not used for analysis.

Once these patterns of activity were identified, we used the data to predict dates of emergence for sensored nests. Our prediction model was generated using only data from *Caretta caretta* nests that met the following conditions (n = 43): 1) were viable, 2) had documented emergence dates, and 3) were not excavated early in order to save hatchlings from impending hurricane/tropical storm surges.

Other factors, such as the length of the hatching or posthatch periods and the impact of nest depth/clutch size on emergence dates were calculated for the same nests, provided nests had available data for nest depth and confirmation of number of hatched eggshells upon excavation. Under normal conditions, excavations occurred at least 72 hours after a boil or 80 days after laying [36]. Both hatched and unhatched eggs were counted at nest excavation. Because turtle eggs do not shatter when the hatchlings emerge and a relatively intact egg casing can still be detected after hatching, accurate hatchling counts can be obtained based on eggshell counts at excavation. Correlations were calculated using Pearson's correlation coefficient.

## TurtleSense visual confirmation

*Lepidochelys olivacea* (Olive Ridley) sea turtle nests in Ostional, Costa Rica were outfitted with TurtleSense in 2018 (n = 5). Video-monitored nests were relocated in trenches set with plexiglass walls and a team from BBC filmed the pipping and hatching from within the nests. The video footage was matched against the motion data to pinpoint nest events, including pipping and activity confirmation.

## Results

### Sensor effectiveness

TurtleSense was successfully able to specifically detect motion within sea turtle nests over the entire course of incubation. Initial data recorded from the newly laid nests showed slow oscillations of the readings at a very low level for many weeks during incubation, consisting of patterns that did not match with tides, temperature, or weather conditions. Measured readings

from a control sensor buried in the sand at the same depth as a typical nest were virtually identical to the sensors in a nearby nest, establishing relative background readings.

We found that the system is sensitive to different types of movement occurring within the nest besides turtle movement. TurtleSense can remotely detect potentially dangerous nest events, such as ocean tidal over-washes and nest predation. These motion events overlay the monitored turtle hatching activity in the accelerometer readings, and so must be carefully interpreted. An undulating sharp discontinuity in the readings of the background noise indicates nest overwash events, which were confirmed in multiple nests by visual inspection. Other nest disturbance events, such as ghost crab predation or perturbations due to NPS nest checks, appear in the data as very large sharp upward spikes in readings that are many times higher than the background noise, most confirmed by visual examination of the nests the subsequent day [Fig 2].

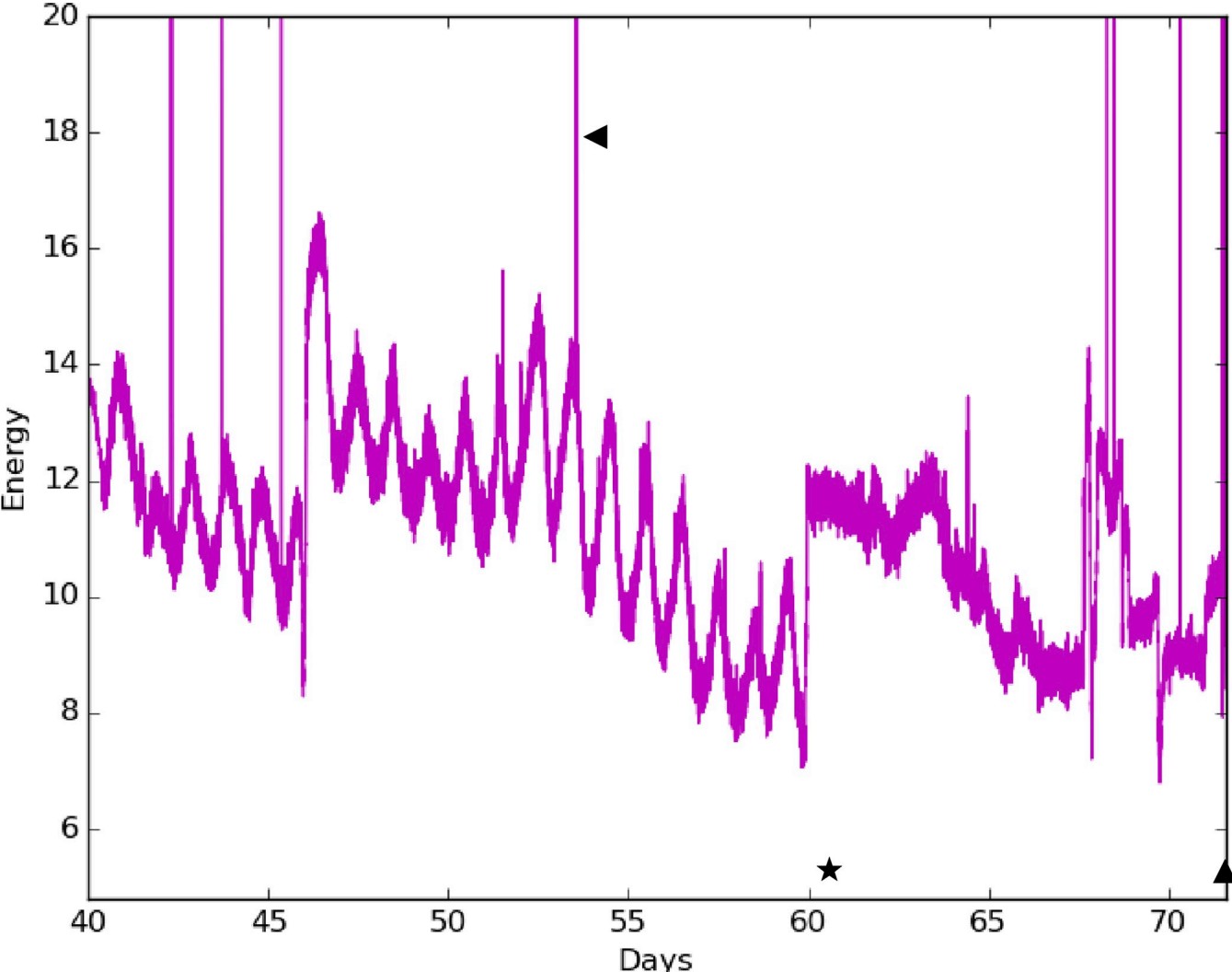

**Fig 2. Energy plot exhibiting overwash and nest disturbance representative motion data.** This infertile in-situ nest was located in an Off Road Vehicle area in Avon, NC (NH013; 2015). Multiple tidal overwash events were recorded at this nest site spanning Days 3–71. The key features are the undulating waveform and the sudden discontinuity in the average background signal. On Day 60, an Off Road Vehicle knocked over the corner sign on the nest enclosure (star), though the nest remained intact. Hatching activity was never detected and the nest was excavated on Day 71 to confirm an infertile nest with 72 undeveloped eggs. Spikes in the data are representative of nest disturbances, such as NPS nest checks or ghost crab predation events (triangles).

## Sea turtle developmental events

The TurtleSense system allows us to remotely detect previously hidden events that occur throughout sea turtle development. Our data analysis shows predictable activity patterns generated inside sea turtle nests during the incubation period, which can be divided into four stages, termed the incubation, prehatch, hatching, posthatch phases.

These activity patterns are organized into 4 key stages:

1. Initial Incubation: After the eggs are laid and the sensor is placed, the early incubation stage is characterized by a long period of inactivity. The initial incubation period lasted between 46–53 days in monitored *Caretta caretta* nests. This period shows little or no motion besides that of background noise.

2. Prehatch: At the conclusion of the initial incubation, there is sometimes a subtle activity decline for a day or more before the hatch period begins, often reaching a new low. This did not appear in all monitored nests (approximately 1/3) and it is unclear what this quieting may indicate.

3. Hatching: The prehatch stage is followed by a period of intense hatching activity. We are accurately able to detect the initiation of hatching activity in the nests and the cessation of hatching activity. Loggerhead sea turtles hatch for an average of 2.22 days +/- 0.180 days (SEM, n = 43; Fig 3).

   During hatching, motion data depict numerous waves of activity in the nest interspersed with quiet interludes (termed "swings"). Data swings are both higher and lower than the slow background noise oscillations, but not nearly as big in magnitude or as sudden as the predation events. The swings in this period of activity are more frequent and clustered than the background noise oscillations, occurring at least 3 times a day or more. These swings stay roughly in the same range and frequency for the several days and are always followed by hatchlings emerging from the nest.

   These activity bursts show synchronized periods of turtle activity and movement cessation that have been confirmed by infrared video footage recorded by the BBC within the nest (See S1 Video). Hatching does not happen simultaneously, but rather it ebbs and flows over a period of several days as the turtles emerge from their eggs.

4. Posthatch: We detect a very inactive period between hatching and emergence that lasts for an average of 1.46 days +/- 0.177 days (SEM, n = 43; Fig 3). The conclusion of the hatching swings is defined by an abrupt stop in the oscillations. The data readings from that point on remain relatively flat or once again look like background noise.

   During this time, the turtles have pipped and emerged from their eggs, but there is very little individual turtle movement in the nest, and there is no coordinated movement. This profoundly quiet pause in activity after hatching was consistently found in all the nests monitored.

## Hatchling emergence

Emergence is initiated in Loggerhead turtle nests 3.62 days +/- 0.23 day (n = 43) after the start of hatching, calculated from the combined duration of the hatch and posthatch phases. The minimum wait time between the start of hatching and first emergence onto the sand was 1.8 days and the maximum was 7.5 days (Fig 3; Table 1). Emergence events are not detectable with the current sensor placement because the turtles are high above the sensor at emergence, but these nests were monitored at night by nest sitters and by NPS each morning for accurate emergence counts.

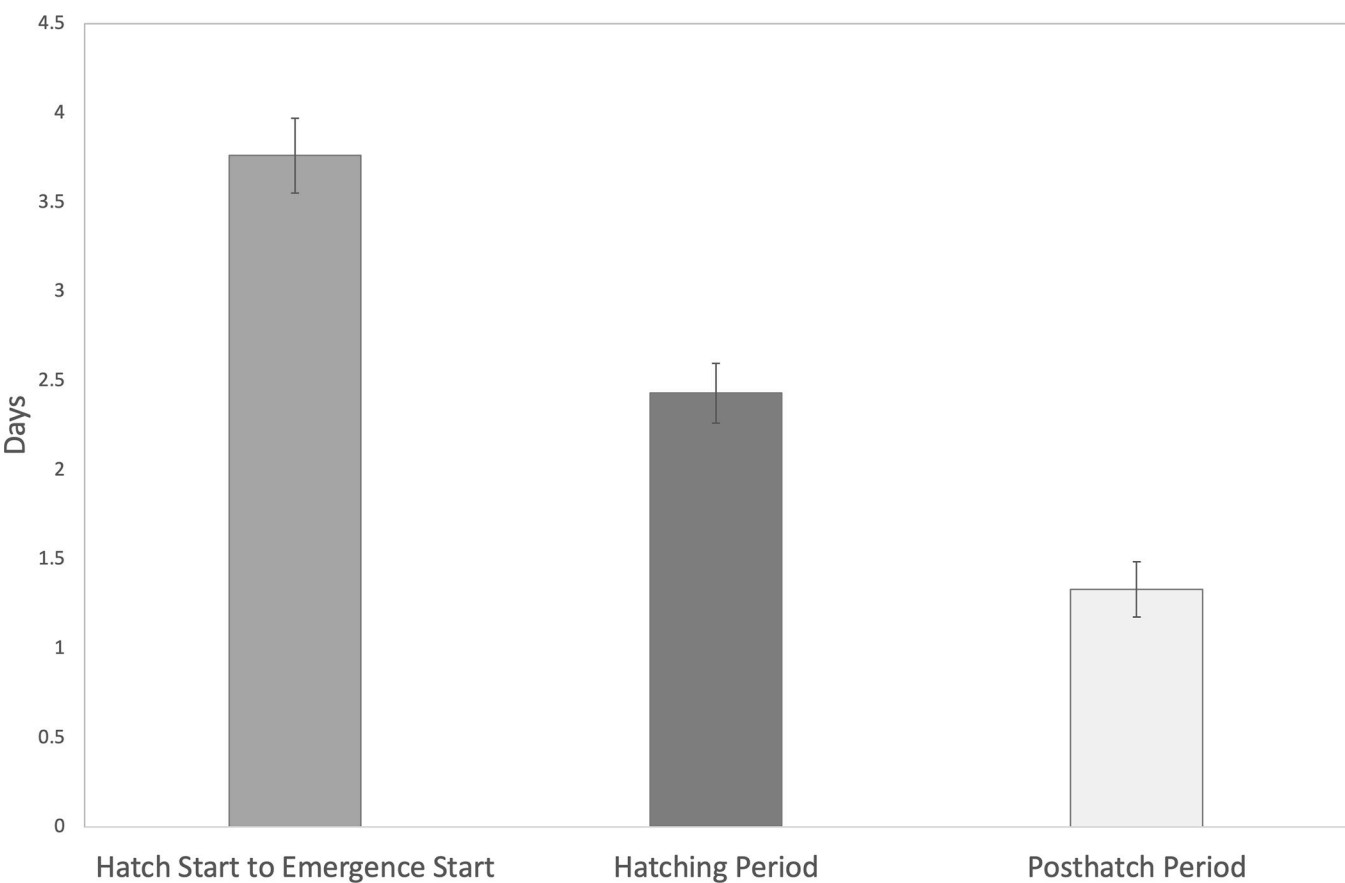

**Fig 3. Timing of hatching and post-hatch phases in sea turtle nests.** Emergence onto the sand is initiated in *Caretta caretta* nests 3.62 days +/- 0.23 day (n = 43) after the start of hatching from the eggs, calculated from the combined duration of the hatch and posthatch phases. The minimum wait time between the start of hatching and first emergence was 1.8 days and the maximum was 7.5 days. Hatching lasts an average of 2.21 days +/- 0.18 days ()n = 43) and the posthatch period lasts for an average of 1.46 days +/- 0.18 days (n = 43). Error bars are SEM.

## Nest predictions

In the absence of sensor data, emergence events are typically predicted by age of incubation and the presence of a depression. However, in our monitored nests, a depression was observed in only 45% of nests (33 out of 74). Based on TurtleSense findings, we used these newly described sea turtle development patterns to create prediction models based on what is happening within the nests. Using these patterns, we were able to predict emergence events, since egg hatching shows up in our data a few days before the hatchlings boil (emerge en masse) or trickle (emerge a few hatchlings at a time, sometimes over several days) out of the nest (Fig 4A and 4C). Importantly, we are also accurately able to accurately predict nest viability. Infertile/

**Table 1. Nest phase lengths and summary statistics for viable *Caretta caretta* TurtleSense monitored nests between 2014–2017 on Hatteras Island, NC.**

| | Nest Phase Lengths (n = 43) | | | Nest Statistics (n = 42) | | |
|---|---|---|---|---|---|---|
| | Hatch phase (days) | Posthatch phase (days) | Start of hatching to emergence (days) | Nest depth (cm) | Total eggs | Number of live hatchlings |
| Mean | 2.21 | 1.46 | 3.62 | 29.73 | 113.6 | 91.57 |
| SEM | ±0.18 | ±0.18 | ±0.23 | ±1.21 | ±3.04 | ±5.03 |
| Range | 0.5–4.1 | 0.5–3.8 | 1.8–7.5 | 15–50 | 80–164 | 6–157 |

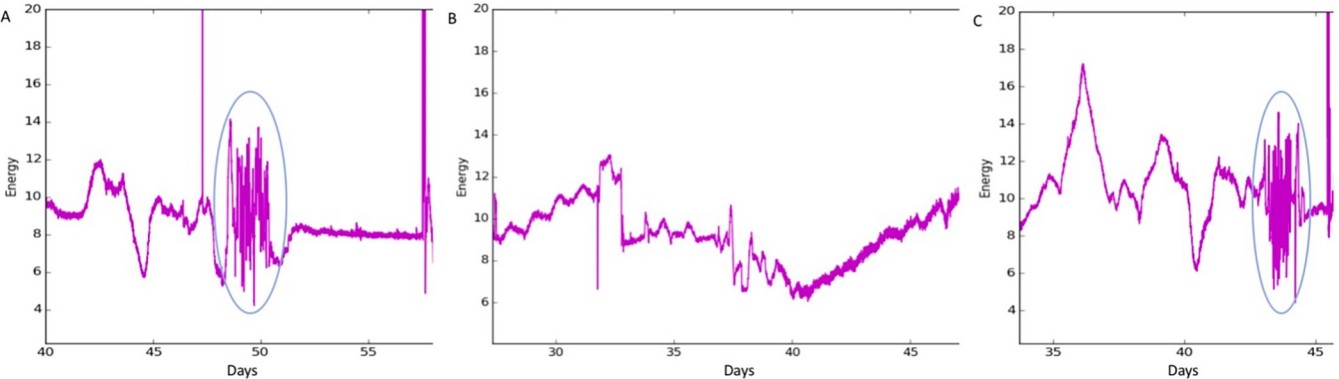

**Fig 4. TurtleSense activity readings for *in situ* fertile and infertile nests during the 2016 nesting season in Cape Hatteras National Seashore, NC.** A) A typical fertile nest has a long initial incubation period, followed by tightly clustered hatching activity and a posthatch period before emergence. TurtleSense detected hatching in this nest from day 47.5–50.5. A depression was observed on Day 50. Emergence was noted as a boil on Day 53, followed by 6 more turtles emerging on Day 56. Nest excavation found 92 hatchlings had successfully left the nest, 5 alive hatchlings, and 7 unhatched eggs (Day 57). Circle indicates hatching activity. B) This infertile nest was overwashed multiple times (Days 31–38) and had a significant amount of sand accumulation on the nest. Sensor data never detected hatching activity and is typical of data seen in a nonviable nest. Early nest excavation due to impending tropical storm Hermine found the nest to be dead with 117 unhatched eggs (Day 47). C) This nest hatched early and so did not yet have nest expansion protections. There was no depression and no turtles had emerged, but sensor activity detected completed hatching activity (Day 43–44). In response to the TurtleSense data and impending tropical storm Hermine, this nest was excavated early (Day 46) to reveal 86 live hatchlings near the surface of the nest cavity, which were subsequently released. There were also 11 unhatched eggs.

unviable nests have little to no detectable motion inside the nest and are typically detected by the absence of hatching activity (Fig 4B).

We established a stepwise prediction method for sea turtle hatchling emergence prediction based on this motion pattern and on the duration of previously recorded hatching and post-hatch phases over the course of several nesting seasons. Our prediction model takes previous average hatch and post-hatch periods into account to predict the start of emergence (Fig 5). This prediction can be first made after the initial hatching day. This prediction method consists of three steps:

1. Observe a full day of hatching. There should be at least 3 or more of these swings within a 24 hour period, often consisting of lower and higher swings than previously recorded.

2. The initial prediction can be made by adding 3.6 days from the start of the observed hatching, remaining cognizant of 1.8 day minimum and 7.5 day maximum observed periods

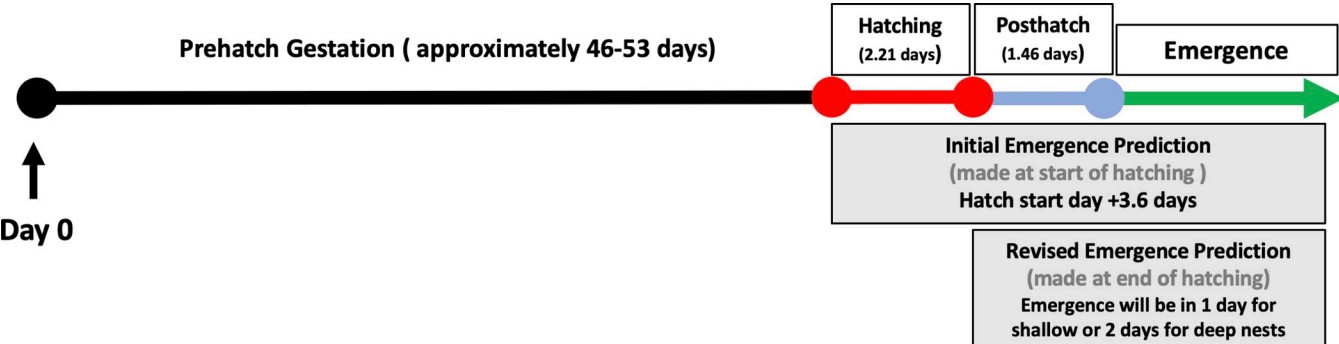

**Fig 5. TurtleSense can be used to accurately predict emergence.** Once egg hatching activity is detected, emergence onto the sand will occur on average in 3.6 days. When the end of hatching is detected, typically turtles will emerge in 1 day from shallow nests (<31 cm) and in 2 days in deep nests (>31 cm), based on a significant correlation between the posthatch period and nest depth.

between hatching and emergence. The initial prediction formula is Emergence = Date of first hatching activity + 3.6 days.

3. Once hatching is completed, the emergence window in nests of unknown depth can be further recalculated period by adding 1.5 days to the date when observed hatching activity ends. If the nest depth is known, we can provide predict hatchling emergence to the day at this point. The narrowed prediction is Emergence = end of hatching activity date + 1 day for nests <31 cm deep or end of hatching activity date + 2 days for nests >31 cm deep.

## Activity confirmation in other sea turtle species

TurtleSense monitored two green turtle (*Chelonia mydas;* n = 2) nests on Hatteras with similar results. In addition, patterns of activity in Hatteras *Caretta caretta* nests were confirmed using video footage over the course of incubation in nests monitored with our sensors in Costa Rica in Olive Ridley (*Lepidochelys olivacea*) nests (n = 5; S1 Video). These data were very similar, but sped up slightly because of a shorter incubation period (observed pipping at 44.5 +/-0.29 days and observed emergence at 47.6 +/- 0.4 days; Fig 6).

## Nest depth and emergence

Nest depth was recorded when the sensor was placed. The mean nest depth was 31.41 cm (+/- 1.22 SEM; n = 43). TurtleSense activity was used to compare both the hatching and posthatch periods to the depth of the nest (Fig 7). We show that there is no correlation between nest depth and the length of time that turtles remain under the sand from hatching initiation until actual emergence ($r(41) = .1814$, $p = .24$, Fig 7A). There is some evidence that nest depth is

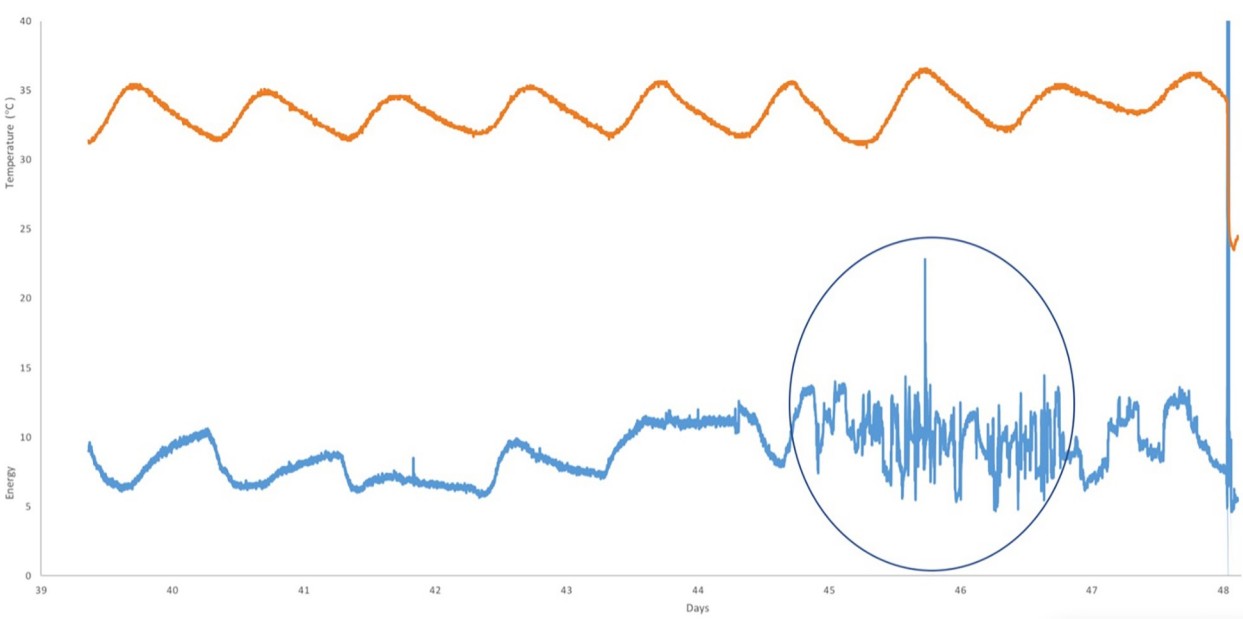

**Fig 6. Detecting hatching events using TurtleSense.** TurtleSense was used to monitor this relocated Olive Ridley (*Lepidochelys olivacea*) nest in Ostional, Costa Rica. The sensor was placed when the nest was relocated on Day 0, and the communication unit was connected to the sensor at Day 39. Temperature data (top line) and accelerometer readings (bottom line) were recorded. Pipping was observed on Day 44 at 1900 via video footage within the nest and clearly matches documented accelerometer data collected by the embedded sensor. In this particular nest, hatching occurred over a 2.1 day period (Day 44.7–46.8; Circle indicates hatching activity). After a period of quiet (approximately 1 day), a trickle emergence was documented late on Day 47 (beginning at 19:20).

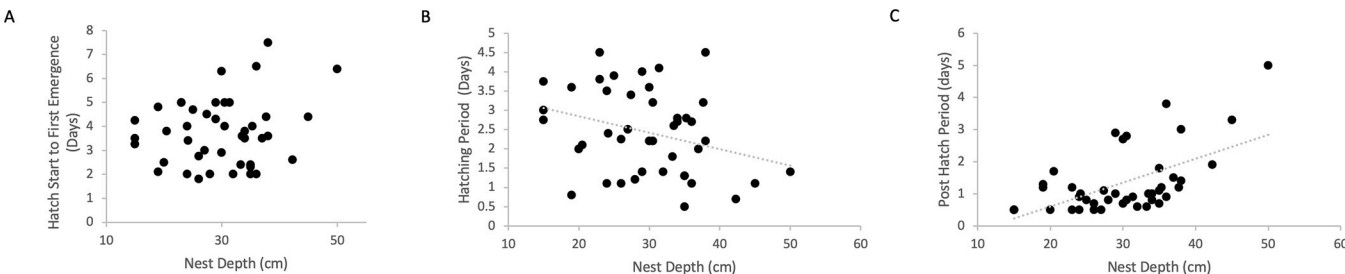

**Fig 7. The impact of nest depth on phase duration.** Deeper nests are correlated with a longer post-hatch phase in *Caretta caretta* hatchlings (n = 43 nests from 2015–17 nesting seasons). A) The time between the start of hatching and the first turtle emergence is not increased in deeper nests ($r(41) = .1814$, $p = .24$). B) There is a weak negative correlation between depth of the nest and hatching period duration ($r(41) = -.3075$, $p = .0449$). C) There is a strong positive correlation between nest depth and time to emergence, where deeper nests have a significantly longer posthatch phase ($r(41) = .5805$, $p = .000045$).

correlated to the length of hatching period ($r(41) = -.3075$, $p = .0449$, Fig 7B). There is a strong correlation between nest depth and the length of the posthatch period, so deeper nests tend to have longer post-hatch phases ($r(41) = .5805$, $p = .000045$, Fig 7C), though there was no motion evidence that the turtles spent more time digging during this time.

## Hatchling number and emergence

TurtleSense activity during hatching and emergence periods were compared to the number of live hatchlings in the nest (average = 91.57 hatchlings per nest +/- 5.03 SEM; n = 43). We found a weak positive correlation between number of live hatchlings present and amount of time between the first hatch and the first emergence ($r(40) = .34$, $p = .0026$, Fig 8A). Increased numbers of hatchlings in the nest do not take longer to complete hatching ($r(40) = .19$, $p = .0232$, Fig 8B), nor is the posthatch period between hatching and emergence significantly different ($r(40) = .257$, $p = .1004$, Fig 8C). No correlation was observed between total number of eggs and the length of hatching, posthatch, or total hatch/posthatch period.

## Discussion

We designed a low-cost, remote way to successfully monitor patterns of motion over the incubation period in *Caretta caretta* nests. Our data show that there are distinct motion phases in the sea turtle hatching and nest emergence process, and that the timing of these events can be used to reliably predict when hatchlings will emerge onto the beach. The system also can

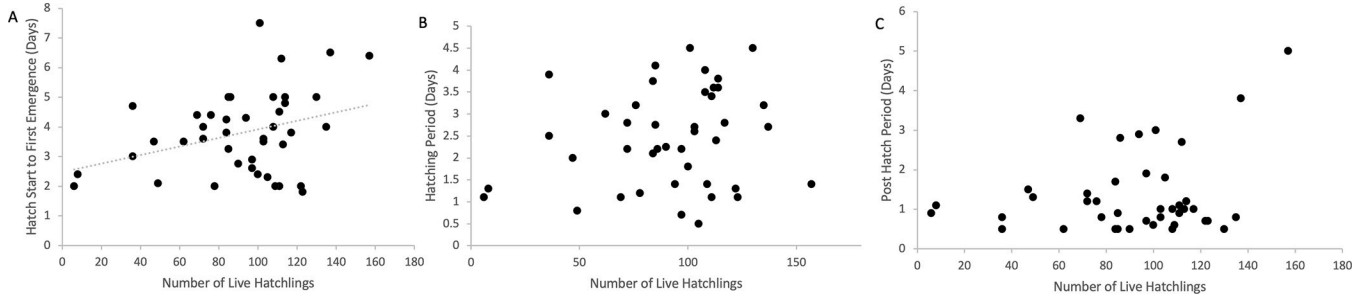

**Fig 8. The impact of number of hatchlings on emergence timing.** There is a weak positive correlation between the number of *Caretta caretta* hatchlings present and how long it takes for the hatchlings to emerge from the nest (n = 42 nests from 2015–17 seasons). A) More nest hatchlings do weakly correlate with increased amount of time between the first hatch and the first turtle emergence ($r(40) = .34$, $p = 0.026$) B) There is no correlation between number of hatchlings and actual hatching period ($r(40) = .19$, $p = 0.232$) and C) There is no correlation between the length of the post-hatch period and the number of hatchlings present ($r(40) = 31$, $p = .084253$), though there is a positive trend.

accurately identify non-viable nests due to the absence of hatching activity. The strength of our predictive model lies in the ability to use real-time data to detect the cessation of hatching activity and prepare for imminent hatchling emergence within the next 1–2 days (based on nest depth, if known).

Our data reveal several synchronous patterns of motion as the result of turtles hatching and then moving around in the nest, but we also detect periods of relative quiet (specifically just before and just after the hatching period). We group these nest motions events into four stages, termed the incubation, prehatch, hatching, posthatch phases.

## Incubation

The first long period of initial incubation appears to show little motion besides that of background noise.

## Prehatch

The second stage, prehatch, is a short period of a day or two when motion actually declines. This may be related to the initial pipping of the eggs. When the hatchlings pip, fluid in the eggs drains out the bottom of the nest, and when the eggs settle, it generates a chamber of air in the nest cavity. The space might be isolating the sensor from some of the background noise, showing quiet readings in a range that is usually lower than most, if not all, of the readings that came before.

## Hatching

The third stage is the majority of hatching activity, which lasts for a few days, reveals numerous waves of motion with very quiet interludes. As each hatchling leaves the egg, it creates a wave of detectable commotion, and then the turtles quickly quiet down, which is supported by the placement of infrared cameras within some nests [37]. Our work demonstrates that hatching activity ebbs and flows over approximately a 2-day period.

## Posthatch

The fourth detectable nest event, termed the posthatch phase, shows a very inactive period that lasts from a few hours to a few days, spanning the time between the end of hatching and emergence onto the sand. On average, this a quiet posthatch/pre-emergence period lasts approximately 1.5 days, and can be modulated by nest depth, though we see no evidence of increased motion during this time that could indicate digging activity.

Video from Olive Ridley nests in Costa Rica also showed that when a hatchling emerged from its shell and made its way to the top of the clutch of eggs, it caused most or all of the hatchlings already assembled to adjust their position to accommodate the new arrival. This assembly at the top of the clutch was where the egg sensor was placed. After the disruption by the new arrival, all the hatchlings stopped moving until the next disruption. This repeated disruption and quieting corresponds to our repeated readings of much greater activity followed by things getting quieter during the hatching period.

Surprisingly, there were very few or no dramatically large motions recorded when actual emergence happened. The sensors were placed just beneath the top 10 eggs in the nest based on the assumption that hatchlings congregate at the top of the nest before a boil. The expectation was that when the hatchling moved past the sensor egg, that motion would cause very high readings. Although this sensor placement allowed for precise detection of hatching activity and the posthatch phase, it was difficult to determine the precise time when boils occur

from collected data. Hatchlings surrounding the sensor could very well be the reason that environmental noises were attenuated. The bodies of hatchlings, being soft and mostly inelastic, would absorb sounds and impacts from their surroundings. The implication of this, though, is that the hatchlings surrounding the sensor are not moving.

It could be that the hatchlings that had already started moving up through the sand were already above the sensor, but observations from the field seem to contradict that theory. Data from one nest, NH047, showed that the erratic period had ended, motion had quieted down, and a boil was imminent. Several nest inspections in the three days before the boil showed that the hatchlings had congregated around the sensor, and all the readings for those last few days were low and undramatic. Our sensor placement was chosen in order to not disturb hatchling emergence, but to also allow the sensor to move down and still detect movement rather than be dangling in the air as hatching activity caused the nest to collapse. Alternative sensor placement or the use of multiple sensors (where a first sensor is placed among the top few eggs and a second sensor just beneath the sand surface) could provide even more information about nest events and potentially detect precisely when the hatchlings climb to the surface to signal a boil in real-time.

Previous findings show that larger groups dig their way out faster than smaller groups in relocated nests [11]. While it is possible that nests with more hatchlings have a longer incubation and prehatch period, our results show that nests with more hatchlings do not have longer hatching periods, nor is the time between egg hatching and emergence onto the sand faster than in nests with fewer hatchlings. In fact, we find a weak positive correlation between more hatchlings and longer hatch to emergence periods, meaning that more hatchlings actually take longer to emerge.

Though previous findings report that the depth of the nest has no effect on the hatching period [17], we observe a weak negative correlation between hatching period and nest depth, where deeper nests take longer to hatch. In addition, we confirm previous findings that the deeper the nest, the longer it takes the hatchlings to emerge once hatching is completed [17]. We also observe a quite strong positive correlation between the posthatch phases and nest depth, though it is important to note that we saw no evidence of more digging activity during this time-period. It is possible that differences exist in hatching and emergence behaviors between in situ and relocated nests, but we did not observe noticeable differences in our activity monitoring.

Future analysis of collected temperature data from the sensors can provide information about whether seasonal temperature fluctuations can impact detected hatching or post-hatch duration periods, as well as monitor precise temperature when emergence occurs. Asynchronous emergence events ("trickles"), rather than large synchronous "boils," can be associated with large within nest temperature ranges [38], so this information could possibly be used to predict boils.

More research is needed to elucidate exactly what other factors can impact the length of the posthatch period, though temperature may be an important variable. We found that the accelerometer readings alone provided ample data to accurately pinpoint emergence dates and we did not need to use the collected temperature information as a predictor of emergence.

We theorize that motion disruptions within the nest allows hatchlings to synchronize nest activity. When incubation nears completion, hatchlings start to hatch at about the same time, but not completely simultaneously, resulting in a small wave of detectable activity after each individual hatching that rises and then settles during the hatching period. As motion ceases within the next, it is possible that this final quieting down might be the cue to the hatchlings that all their siblings have hatched and it is time to leave the nest. The ability of hatchlings to sense a sufficiently long period of calmness after hatching and a drop in temperature might be

what triggers a boil. These two simple biological inputs might be all that is necessary for the instinctual programming that synchronizes the behavior of all the hatchlings.

Our results also call for a careful investigation of vibration/motion-based sea turtle sibling communication. Vibration cues for hatching have been isolated in other reptiles, including the pig-nosed turtle (*Carettochelys insculpta*) [39], where groups of eggs hatch faster than single eggs treated in the same way, indicating a function of sibling vibrations [40]. The length of time between the movements within vibrational cues can signal embryos within a clutch to hatch, as seen in predator-induced early hatching of red-eyed treefrogs (*Agalychnis callidryas*), where vibrations were enough to induce early hatching and embryos were able to differentiate between different vibration stimuli [41]. Future research could combine vibration data collection with vocalization monitoring to generate a more complete picture of embryo-to-embryo communication.

Because sea turtles are a threatened species, each municipality or locality must adopt a nest management strategy, most of which are labor intensive and relatively inaccurate at predicting day of emergence. As an automated system, TurtleSense can accurately predict hatching events and send alerts to wildlife managers and researchers. This is particularly important, since depressions, the standard way of identifying a hatching nest, were present in less than half of our monitored nests. In addition, current protocols mandate that protections be given for all nests, despite the fact that hatching success between 2009–2014 ranged from 33–77% in sea turtle nests (depending on month laid and if they were relocated) in Cape Hatteras National Seashore [42]. TurtleSense can accurately predict infertile nests and eliminate the unnecessary funneling of resources towards their protection. TurtleSense can also provide information to remotely notify conservation teams about nest dangers posed on beaches in real-time, including predation, potential poaching activity, or tidal inundation, and allow management to make decisions about whether a nest contains hatchlings instead of eggs that now need to be rescued from storm surges.

TurtleSense has the ability to transform sea turtle nest management, providing useful real-time data to enable more efficient utilization of resources, better hatchling protection, and fewer volunteer hours. Accurately pinpointing emergence events allows for "just in time" barrier erection, reducing the impact of turtle management induced beach closures on beach goers, tourism, and the fishing industry. We primarily examined *Caretta caretta* nests, though we did also monitor pilots in Olive Ridley and Green Turtle nests. We saw the same basic patterns in these nests, so we anticipate an easy transfer of this predictive model to other turtle species. This was the case in Costa Rica, where we were able to detect and predict Olive Ridley hatchling emergence accurately, even though all nests were relocated and we had not previously used TurtleSense in this species.

TurtleSense seeks to both preserve the health of endangered sea turtle hatchlings and lessen the impact of beach closures for people, freeing up valuable volunteer resources. Annually, in the state of North Carolina, the NPS spends $2.0 million dollars to support coordination of more than 600,000 volunteers statewide [43]. Specifically on Hatteras Island, nest sitting includes the following tasks: arriving at dusk to clear any off road vehicle tracks in the sand and sitting with the nest from 7:30pm to 12:00am. Generally, crowds can form around the sea turtle nests located near villages. In these situations, nest sitters are required to maintain crowd control, while also ensuring the nest stays safe during emergence. Nest sitters typically work in pairs assigned to one nest, where they wait for emergence each night (approximately 5 hours) ranging from 4–8 days for each nest beginning on a specific date after the nest was laid. We estimate that this protocol of 2 volunteers for 4–8 days per nest requires approximately 40–80 volunteer hours per nest.

On Hatteras Island, nest sitters typically only begin sitting on a nest when a depression has formed, the nest is beginning to be within the average emergence day for the season, or other clear indicators that emergence is imminent. TurtleSense can instead use motion data to provide a clear signal if the volunteers need to begin nest sitting based on when hatching is completed. Through the use of TurtleSense, the end of hatching activity would signal the need for nest sitters the following day for shallow nests, and two days later for deep nests. This potentially reduces the volunteer hours to 10–20 hours for each monitored nest, based on 1–2 days of nest sitting for 2 volunteers. Some turtle may still emerge after this date, but the volunteer resources would be "just-in-time" and importantly, TurtleSense would eliminate the need for volunteers to sit with nests that have not hatched yet or that are nonviable.

TurtleSense provides more predictable opportunities to engage in public outreach for emergence events from nests, which is important for both conservation and awareness efforts, as turtle-based ecotourism has positive indirect impacts on the sea turtle population sustainability and conservation efforts [44]. TurtleSense can decrease the number of volunteer hours needed, benefit the park service by providing tighter windows for emergence events, and aid fishermen with shorter beach closures. Although TurtleSense is a useful tool for many reasons, it does not mitigate the need for other turtle conservation tactics. Initiatives such as turtle-friendly lighting should be encouraged or enforced so that interference to nests can be reduced in general, lessening the need for heavy nest management tactics and the potential for misoriented hatchlings.

## Supporting information

**S1 Video.**
(MP4)

## Acknowledgments

We thank the numerous individuals who collaborated on this research over the last ten years, generously donating time, labor, ideas, and resources. These people include Britta Muiznieks, Mark Dowdle, Matthew Godfrey, William Thompson, Amy Thompson, Charles Wade, Christopher Romero, Mark Salvatore, James Washer, Robert "Trebor" Harris, Eric Frey, Lou Browning. We also are grateful to the Hatteras Island Ocean Center, Nerds without Borders, IBM, Jo Young at Renegade Pictures (UK), Janus Remote Communications, Screaming Circuits, Telit, Frontline Test Equipment Inc., Hunan Corun New Energy Co. LTD, and Remee Wire & Cable. We thank the Hatteras Island National Park Service employees for the accommodation of this project and for their dedication to protecting sea turtle nests.

## Author Contributions

**Conceptualization:** Eric Kaplan, Thomas Zimmerman, Samuel Wantman.

**Data curation:** Erin B. D. Clabough, David Hermeyer, Samuel Wantman.

**Formal analysis:** Erin B. D. Clabough, Joshua Chamberlin, Samuel Wantman.

**Funding acquisition:** Eric Kaplan.

**Investigation:** Erin B. D. Clabough, Eric Kaplan, Thomas Zimmerman, Joshua Chamberlin, Samuel Wantman.

**Methodology:** Erin B. D. Clabough, Joshua Chamberlin, Samuel Wantman.

**Project administration:** Eric Kaplan.

**Resources:** Eric Kaplan, David Hermeyer, Samuel Wantman.

**Software:** David Hermeyer, Thomas Zimmerman, Samuel Wantman.

**Supervision:** Erin B. D. Clabough, Eric Kaplan.

**Validation:** Erin B. D. Clabough, David Hermeyer, Samuel Wantman.

**Visualization:** David Hermeyer, Thomas Zimmerman, Samuel Wantman.

**Writing – original draft:** Erin B. D. Clabough, Joshua Chamberlin, Samuel Wantman.

**Writing – review & editing:** Erin B. D. Clabough, David Hermeyer, Samuel Wantman.

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
