## [Decision Letter · Decision Letter 0]

31 Jan 2022

PONE-D-21-39727The secret life of baby turtles: A novel system to predict hatchling emergence, detect infertile nests, and remotely monitor sea turtle nest eventsPLOS ONE

Dear Dr. Clabough,

Thank you for submitting your manuscript to PLOS ONE. This paper has been assessed by 2 subject experts and the Academic Editor; all 3 agree that the work is potentially a good contribution. However, the manuscript as submitted does not fully meet PLOS ONE’s publication criteria as it currently stands. Therefore, we invite you to submit a revised version of the manuscript that addresses the points raised during the review process.

**Required Revisions**

1. Please revise the Introduction to include some description and appropriate literature cited in the area of sea turtle conservation and management. The tool being described would not be necessary without these broader concerns.

2. Results: this section of the manuscript is written in an unusual style for a scientific paper, and all 3 of us had difficulty following what was going on at times, and clearly understanding the application of statistical analyses. The subject expert reviewers provide more detailed comments to help the authors revise this section and ensure that it is correct and clear. 3. Data availability: the authors have not made their data available and suggest that they should be contacted as a single source for the information. This is not an acceptable means of data accessibility according to PlosOne guidelines.

4. Product availability: the Turtle Sense device is likely to be of interest to other researchers in the field; however, there is no information provided on how to acquire them. A brief Google search revealed some information on the study, but not how to acquire devices. The authors should provide some information on this for readers.

**Suggested Revisions** Please carefully consider and address the comments provided by the subject experts.

We look forward to receiving your revised manuscript.

Kind regards,

Christopher M. Somers

Academic Editor

PLOS ONE

Journal Requirements:

2. Thank you for your  interest in the Remote Assessment Call For Papers. After careful consideration, we have decided that your manuscript is not in scope for this Call For Papers. This call for papers invites research on remote assessment on the biomedical fields. We understand that you are presenting an interesting tool to accurately predict sea turtle hatchling emergence. Whilst we appreciate the value to the field, we feel that your submission does not justify inclusion in this call for papers.

Reviewers' comments:

Reviewer's Responses to Questions

**Comments to the Author**

1. Is the manuscript technically sound, and do the data support the conclusions?

Reviewer #1: Partly

Reviewer #2: Partly

2. Has the statistical analysis been performed appropriately and rigorously? 

Reviewer #1: Yes

Reviewer #2: Yes

3. Have the authors made all data underlying the findings in their manuscript fully available?

Reviewer #1: No

Reviewer #2: No

4. Is the manuscript presented in an intelligible fashion and written in standard English?

Reviewer #1: Yes

Reviewer #2: Yes

5. Review Comments to the Author

Reviewer #1: This is an interesting manuscript describing the application of novel technology to a problem in sea turtle management. While I am convinced that the technology described could prove to be useful, the manuscript, especially the Results, which does not conform to the Results section of a typical scientific manuscript, needs work. My specific comments follow:

Line 100: delete ‘emergence’

Line 111-112: citations?

Line 122-133: Here or elsewhere in the introduction the authors should provide information on the conservation status of sea turtles and explain the role of nest monitoring in sea turtle management. Perhaps this should constitute the first paragraph of the manuscript followed by the review of what is known about sea turtle emergence (lines 67-120). After all, the manuscript is more about management strategies than it is about cues for emergence.

Lines 156-178: It seems like citation 19 should be referenced throughout these sections. I found the figures in that citation to be exceedingly helpful in understanding the design of the TurtleSense system.

Line 211: delete second ‘instead’

Line 194-233: a flow chart might be helpful.

Lines 252-256: this information seems out of place here.

Lines 296-301: more information needs to be provided on how ‘periods of activity’ were identified. How were they distinguished from background? How were the onset and end of such periods defined? Was there ambiguity in some recordings (not captured in the figures)?

Lines 306-307: some explanation of how hatched eggshells are counted should be provided. Given the large number of eggs per nest, can this be done accurately? Do shells remain as a single piece or do they get fragmented?

Lines 310-315, 418-421: an image captured from these videos would be and interesting figure!

Lines 353: “there is there is”

Lines 314-466, Results: I found the results section of this manuscript to be non-traditional for a scientific paper in that it lacked detailed results and included material that more typically belongs in the Discussion. For example, lines 320-326 consist of a series of assertions without reference to specific observations to back them up. Similarly, lines 334-354 describe four activity pattern stages but do not provide objective criteria by which one stage is distinguished from the next. A table showing means, standard deviations, and ranges of the duration of these stages (as well as nest depths and hatchling numbers) should be provided. Because of ambiguity in how these periods are determined, the patterns revealed by correlational analyses (Fig 5 and 6) are difficult to interpret. It may also be difficult for managers of other sea turtle populations at other beaches to use the nest prediction steps described in lines 372-416, thus limiting the utility of the approach described here.

Lines 379-380 and elsewhere: I am confused by the terms ‘popping’, ‘boil’, and ‘trickle’; they should be defined.

Line 436: a citation should be provided regarding what previous investigators found and this should be moved to the introduction. If possible, a priori predictions regarding nest depth and number of hatchlings should be made explicit in the introduction. The interpretation of these analyses (e.g., line 459) should be saved for the discussion.

Lines 496-502: Can arrows or other symbols be added to Fig. 4 corresponding to the description provided here? Why not include the video and trace as supplemental material?

Lines 504-513: does this suggest that a different sensor placement should be considered?

Lines 540-545: why not include those analyses here?

Lines 547-557: the popcorn metaphor seems to be a bit of a stretch. Popcorn kernels are not responding to what other kernels are doing – but you suggest that turtles do.

Lines 570-590: to what degree can the predictive model described here be used elsewhere? Will it take multiple hatching seasons to recalibrate the predictive model for each turtle species and location? It would be helpful to provide information on cost and effort necessary to deploy TurtleSense. How does the cost and effort compare to current monitoring methods. Funds for management are always limited!

Lines 593-664: reference formatting needs careful editing.

Line 606-607: citation is incomplete

Fig. 5 and 6: Statistical analyses consisted of tests for significant correlations but the figures show (presumably) regression lines, even in cases when the correlation was non-significant. While the lines allow visualization of the correlation, they are typically omitted in the case of non-significance. Alternatively, if nest depth and number of hatchlings are thought to have a causal effect on period length, why not use regression from the outset? Then the magnitude of the effect (number of days in for shallow vs. deep nests) could be included in the results.

Other: I did not see information on funding sources or Institutional Animal Use and Care approval.

Reviewer #2: Summary -

The authors use a novel system to detect movement inside sea turtle nests which can signal activity levels at various stages in the nest. They show it is possible to detect when nests are non-viable and can ascertain the timing of hatching from eggs. However, the technology failed to determine when hatchlings are crawling up and about to emergence from the sand, but the authors suggest further changes in the study methods could improve this flaw. They explore the process of mass emergence and report their findings that hatching from eggs occurs over a period of a couple of days and that movement within the nest may play an important role in hatchling communications. They consider factors such as nest depth and number of hatchlings which could influence the timing of the hatching and emergence stages and report a way of predicting emergence based on their data. This is an interesting study which other sea turtle conservation programmes may find useful, but the paper requires major revisions as I found many discrepancies, some of which just require better clarification, for it to fit into the requirements of the journal

Introduction

General comments: Literature cited is limited throughout. For example, a key reason to use this technology is to prevent misorientations from artificial lighting, but there is very little explanation of this and its severity for the reader to understand - there are many studies on this particular topic which should be cited. Check font - sometimes inconsistent.

Line 119: It is mentioned that the device is 'wireless' yet in the methods it states that it needs to be attached to a 6-meter cable that plugs into the communication unit. Please correct or clarify.

Lines 118 - 120:This final sentence should be at the end of the introduction, where you introduce TurtleSense.

Line 143: ...enable the erection 'of' "just-in-time" barriers.

Methods

General comments: It would be useful for the reader to know the approximate cost for the materials to build one unit, especially as it is said to be cost effective. In addition, does the unit itself (that is placed into the nest) give off any heat / have potential to impact hatch success? It would be useful to compare hatches of nests with TurtleSense to those without (removing nests that were relocated or had impacts from inundation/erosion or other threats).

Line 185 - 186: The sensor sends data to the commications unit when it is ready to upload a report to the internet - It isn't clear what this is based on, is it when the storage reaches a certain limit or is it at regular time intervals?

Line 290: "motion readings from 79 nests between 2014-2017.." but on line 236 it states that 74 nests were monitored with TurtleSense between 2013-2017. Are these different nests? please clarify.

Line 291 - 292: Please add in sample sizes for how many nests were relocated and how many were in-situ. There is also no mention of whether the differences in activities were tested between relocated nests and in-situ nests which is important. Methods on how nest relocations were carried out should be included (was the depth and shape of the natural chamber taken into account etc).

Line 291 - 297: It is mentioned that data from 79 nests are analysed. 72 are loggerhead and 2 are green, what about the other 5? Is it the olive ridley nests that are mentioned below? clarify.

Line 296: More detail needed here on what data were analysed, was it just motion or did this include temperature/other measures?

Results

General comments: It is not always immediately clear when talking about emergence if the authors mean emergence from eggs or emergence from the sand. Please make this clearer throughout, including in Figure 1. Was every infertile nest detected before excavating? How many times was the model correct for predicting the emergence date (from the sand)?

Fig.1: Needs to be higher quality with larger font used.

Lines 373 - 374: In methods it should be added that nests were checked (and how frequently) and that depressions were noted.

Line 376: Repetitive...reword.

Line 379 - 380: This is the first time introducing 'popping' and 'boiling'. Introduce these terms in the introduction if they are to be used as not all readers will understand.

Fig. 2: Day 47 seems too early to be excavating eggs, given hatching was ccurring between 47.5-50.7 days. What was the reason to excavate at this point?

Line 402: Explain what 'swings' are.

Fig.3: Needs to be clearer on the diagram that the +1 and +2 days are used instead of, not as well as, the original 1.5 day estimation.

Line 434: How was nest depth measured? Was it depth to the base of the chamber or the top? Should be in methods.

Discussion

General comments: It should be mentioned that although this is a useful tool for many reasons, other initiatives, such as turtle friendly lighting' should be encouraged (or enforced as many ) so that interference to nests can be reduced in general, rather than allowing the issue to persist and increase. Terminology is switched between 'baby turtles', 'hatchlings' and 'turtles'. Best to choose one - I prefer 'hatchlings' as a more scientific term to baby turtles.

Line 472: I somewhat disagree that this is reliable estimate of when hatchlings are emerging onto the beach as it could be up to 7.5 days. It is providing more of a reliable estimate than previous methods but is still not a method that could be relied upon really for an exact emergence.

Line 525: It needs to be noted that there are likely differences in relocated and in-situ nests as relocated nests will not entirely reflect natural conditions.

Line 576: This is incorrect based on the report referenced. The report states that the average hatch success ranged from 33 - 77% (based on the month) with lower success being linked to storm erosion and nest inundation (which is different to nest infertility). The report referenced does not mention nest infertility.

6. PLOS authors have the option to publish the peer review history of their article (what does this mean?). If published, this will include your full peer review and any attached files.

Reviewer #1: No

Reviewer #2: No

---

## [Author Response · Author response to Decision Letter 0]

4 Apr 2022

March 27, 2022

Dear Reviewers,

We thank you for your thoughtful review of this manuscript. We have addressed your comments below. 

Sincerely,

Erin Clabough, PhD

Reviewer #1:

This is an interesting manuscript describing the application of novel technology to a problem in sea turtle management. While I am convinced that the technology described could prove to be useful, the manuscript, especially the Results, which does not conform to the Results section of a typical scientific manuscript, needs work. My specific comments follow:

Line 100: delete ‘emergence’

AUTHOR RESPONSE: This is now corrected. 

Line 111-112: citations?

AUTHOR RESPONSE: This statement now contains citations.

Line 122-133: Here or elsewhere in the introduction the authors should provide information on the conservation status of sea turtles and explain the role of nest monitoring in sea turtle management. Perhaps this should constitute the first paragraph of the manuscript followed by the review of what is known about sea turtle emergence (lines 67-120). After all, the manuscript is more about management strategies than it is about cues for emergence.

AUTHOR RESPONSE: The introduction has been revised to include this important information. Information has specifically been added to a new information about conservation of sea turtles and nest management with citations.

Lines 156-178: It seems like citation 19 should be referenced throughout these sections. I found the figures in that citation to be exceedingly helpful in understanding the design of the TurtleSense system.

AUTHOR RESPONSE: This citation has now been included in the beginning of the TurtleSense hardware subsection in the Materials and Methods section.

Line 211: delete second ‘instead’

AUTHOR RESPONSE: This is now corrected. 

Line 194-233: a flow chart might be helpful.

AUTHOR RESPONSE: We agree and we have created a flowchart to depict how the accelerometer data is handled and included it as a new figure (now Fig 1).

Lines 252-256: this information seems out of place here.

AUTHOR RESPONSE: This protocol approval information was moved up to the beginning of the TurtleSense field testing section.

Lines 296-301: more information needs to be provided on how ‘periods of activity’ were identified. How were they distinguished from background? How were the onset and end of such periods defined? Was there ambiguity in some recordings (not captured in the figures)?

AUTHOR RESPONSE: We have updated the results section to include specific details about how the accelerometer readings were interpreted. We have also included a new figure that illustrates what various external nest events look like, as TurtleSense can detect these as well. This is now Figure 2

In summary, early on during the first in-the-nest trials of the Turtle Sense technology, the developers had no idea what the data would look like. The first nests started recording data early on, starting at day one or day two. For many weeks the data showed slow oscillations of the readings at a very low level. Because we did not know what these oscillations were, we tried to correlate them (unsuccessfully) with tides, temperature and weather conditions. We wondered if the oscillations might be related to the developing eggs. To test that theory, a sensor was buried in the sand at the same depth of a typical nest, in a location where we knew that there was no nest. The readings we measured were virtually identical to the sensors in a nearby nest, so it was assumed that the readings were the result of background noise. The source of the noise might be a combination of temperature, surf noise, beach traffic, blowing sand, and weather. The actual source was determined to be unimportant from that point on. What became important were readings that did not fit the gentle pattern of those oscillations. 

There were several new data patterns that emerged over time. One pattern was a sharp sudden discontinuity in the readings of the background noise. Over time we were able to correlate these with nest overwashes. The sudden addition of water to the sand around the sensor may have changed the acoustic properties of the sand causing the discontinuity.

Another new pattern was data that showed very large spikes in readings – many times higher than the background noise – for a short amount of time and occurring at random times during gestation. These were frequently correlated with predation events. These predatory events by animals such as ghost crabs could be confirmed in most cases by visual examination of the nests the subsequent day. We also noticed similar readings after nest checks by members of the National Park’s Turtle Patrol. 

With the very first nest we kept looking for data that would indicate nest hatching activity after 40 days of gestation. We quickly identified periods of activity that looked nothing like the other patterns we had seen. These “periods of activity” were identified by data swings that were both higher and lower than the slow background noise oscillations, but not nearly as big in magnitude or as sudden as the predation events. The oscillations were more frequent than the background noise oscillations, occurring at least 3 times a day or more. These oscillations stayed roughly in the same range and frequency for the several days they were noticed. From our first nest we correlated this activity with hatching activity, since it was always followed by hatchlings emerging from the nest a few days later.

The onset of a period of activity was often preceded by a slow drop in the readings. This was not noticed in all of the nests, and we don’t know what this might indicate, if anything.

The conclusion of a period of activity was defined by an abrupt stop in the oscillations. The data readings from that point on remained relatively flat or once again looked like background noise. No patterns were noticed in the data from that point until the hatchlings emerged.

Lines 306-307: some explanation of how hatched eggshells are counted should be provided. Given the large number of eggs per nest, can this be done accurately? Do shells remain as a single piece or do they get fragmented?

AUTHOR RESPONSE: Both hatched and unhatched eggs were counted at nest excavation. The eggs do not shatter, but typically a relatively intact egg casing can still be detected after hatching, which makes counting successful hatchlings possible based on eggshells. This information has been added to enhance information originally provided in Lines 306-307.

Lines 310-315, 418-421: an image captured from these videos would be and interesting figure!

AUTHOR RESPONSE: We agree that this would make an interesting figure. We have provided a video clip as Supplemental Video 1 to better illustrate the visual confirmation techniques. The supplemental video will also visually address the shape and texture of the eggs for the reader.

Lines 353: “there is there is”

AUTHOR RESPONSE: This is now corrected. 

Lines 314-466, Results: I found the results section of this manuscript to be non-traditional for a scientific paper in that it lacked detailed results and included material that more typically belongs in the Discussion. For example, lines 320-326 consist of a series of assertions without reference to specific observations to back them up. Similarly, lines 334-354 describe four activity pattern stages but do not provide objective criteria by which one stage is distinguished from the next. A table showing means, standard deviations, and ranges of the duration of these stages (as well as nest depths and hatchling numbers) should be provided. Because of ambiguity in how these periods are determined, the patterns revealed by correlational analyses (Fig 5 and 6) are difficult to interpret. It may also be difficult for managers of other sea turtle populations at other beaches to use the nest prediction steps described in lines 372-416, thus limiting the utility of the approach described here.

lines 320-326 consist of a series of assertions without reference to specific observations to back them up. 

AUTHOR RESPONSE: We revised this section to remove some of these assertions and provide more information to support the types of motion events the sensors are able to detect. 

lines 334-354 describe four activity pattern stages but do not provide objective criteria by which one stage is distinguished from the next. A table showing means, standard deviations, and ranges of the duration of these stages (as well as nest depths and hatchling numbers) should be provided.

AUTHOR RESPONSE: We have updated the manuscript to eliminate ambiguity about what sorts of patterns are detectable in the data to characterize these different periods and how we sorted that into stages. We have also inserted a table including specifics about this information in our observed nests as a new Table 1.

Lines 379-380 and elsewhere: I am confused by the terms ‘popping’, ‘boil’, and ‘trickle’; they should be defined.

AUTHOR RESPONSE: “Popping” has been replaced as a term by “hatching” in line 379. We have defined “boil” and “trickle” the first time they are mentioned, housed under the section “Nest Predictions.”

Line 436: a citation should be provided regarding what previous investigators found and this should be moved to the introduction.If possible, a priori predictions regarding nest depth and number of hatchlings should be made explicit in the introduction.

AUTHOR RESPONSE: We cannot include a priori information in the introduction because we did not make predictions, but the introduction includes information about what previous research has found regarding nest depth, with citations.

The interpretation of these analyses (e.g., line 459) should be saved for the discussion.

AUTHOR RESPONSE: Interpretation of these analyses (including line 459) was moved to the discussion.

Lines 496-502: Can arrows or other symbols be added to Fig. 4 corresponding to the description provided here? Why not include the video and trace as supplemental material?

AUTHOR RESPONSE: Yes, we agree that a visual is a wonderful addition to the manuscript. We did include a new supplemental video that displays the oscillations in the data more clearly. 

Lines 504-513: does this suggest that a different sensor placement should be considered?

AUTHOR RESPONSE: Yes, certainly that could be considered. Our sensor placement was chosen in order to not disturb hatchling emergence, but to also allow the sensor to move down and still detect movement rather than be dangling in the air as hatching activity caused the nest to collapse. Different sensor placement could be considered, as can the use of more than one sensor per nest. This section has been updated to reflect that information.

Lines 540-545: why not include those analyses here?

AUTHOR RESPONSE: We found that the accelerometer readings alone provided ample data to accurately predict emergence and we did not need to use the collected temperature information. As such, we did not analyze temperature fluctuations as a predictor of emergence, though we did collect the data and it is available for analysis if another research group is interested in temperature. The manuscript has been updated to explain that in the discussion. 

Lines 547-557: the popcorn metaphor seems to be a bit of a stretch. Popcorn kernels are not responding to what other kernels are doing – but you suggest that turtles do.

AUTHOR RESPONSE: We agree that the removal of this metaphor makes the data more clear. The popcorn references have been taken out of the discussion.

Lines 570-590: to what degree can the predictive model described here be used elsewhere? Will it take multiple hatching seasons to recalibrate the predictive model for each turtle species and location?

AUTHOR RESPONSE: We primarily examined Caretta caretta nests, though we did also monitor pilots in Olive Ridley and Green Turtle nests. We saw the same basic patterns in these nests, so we anticipate an easy transfer of this predictive model to other turtle species. This was the case in Costa Rica, where we were able to detect and predict Olive Ridley hatchling emergence accurately, even though all nests were relocated and we had not previously used TurtleSense in this species. This information has been added to the discussion.

It would be helpful to provide information on cost and effort necessary to deploy TurtleSense. How does the cost and effort compare to current monitoring methods. Funds for management are always limited!

AUTHOR RESPONSE: We have added cost of construction information to the Methods section under Hardware, and we have added information about the comparative cost of volunteer hours to the Discussion section.

Lines 593-664: reference formatting needs careful editing.

AUTHOR RESPONSE: The reference formatting has been edited to conform to Vancouver style.

Line 606-607: citation is incomplete

AUTHOR RESPONSE: This citation is now corrected.

Fig. 5 and 6: Statistical analyses consisted of tests for significant correlations but the figures show (presumably) regression lines, even in cases when the correlation was non-significant. While the lines allow visualization of the correlation, they are typically omitted in the case of non-significance. Alternatively, if nest depth and number of hatchlings are thought to have a causal effect on period length, why not use regression from the outset? Then the magnitude of the effect (number of days in for shallow vs. deep nests) could be included in the results.

AUTHOR RESPONSE: The regression lines have been removed from the last 2 figures for nonsignificant results. The regression analyses were performed and the correlation coefficient (r) is reported as a measure of effect size. 

Other: I did not see information on funding sources or Institutional Animal Use and Care approval.

AUTHOR RESPONSE: The animal study was reviewed and approved by the National Park Service. NPS permit numbers updated in the Methods section. This project was not grant-supported, but was funded by volunteer work and supported by agreements between the Hatteras Island Ocean Center nonprofit and the National Park Service. 

Reviewer #2: 

Summary -

The authors use a novel system to detect movement inside sea turtle nests which can signal activity levels at various stages in the nest. They show it is possible to detect when nests are non-viable and can ascertain the timing of hatching from eggs. However, the technology failed to determine when hatchlings are crawling up and about to emergence from the sand, but the authors suggest further changes in the study methods could improve this flaw. They explore the process of mass emergence and report their findings that hatching from eggs occurs over a period of a couple of days and that movement within the nest may play an important role in hatchling communications. They consider factors such as nest depth and number of hatchlings which could influence the timing of the hatching and emergence stages and report a way of predicting emergence based on their data. This is an interesting study which other sea turtle conservation programmes may find useful, but the paper requires major revisions as I found many discrepancies, some of which just require better clarification, for it to fit into the requirements of the journal

Introduction

General comments: Literature cited is limited throughout. For example, a key reason to use this technology is to prevent misorientations from artificial lighting, but there is very little explanation of this and its severity for the reader to understand - there are many studies on this particular topic which should be cited. Check font - sometimes inconsistent.

AUTHOR RESPONSE: The introduction has been revised to include this important discussion about current reasons for good nest management, including artificial lights (with citations).

Line 119: It is mentioned that the device is 'wireless' yet in the methods it states that it needs to be attached to a 6-meter cable that plugs into the communication unit. Please correct or clarify.

AUTHOR RESPONSE: The system is wireless but the sensor itself is not. The term “wireless” has been removed from this particular sentence to eliminate confusion.

Lines 118 - 120:This final sentence should be at the end of the introduction, where you introduce TurtleSense.

AUTHOR RESPONSE: This sentence has been moved closer to the end of the introduction. 

Line 143: ...enable the erection 'of' "just-in-time" barriers.

AUTHOR RESPONSE: This is now corrected. 

Methods

General comments: It would be useful for the reader to know the approximate cost for the materials to build one unit, especially as it is said to be cost effective. In addition, does the unit itself (that is placed into the nest) give off any heat / have potential to impact hatch success? It would be useful to compare hatches of nests with TurtleSense to those without (removing nests that were relocated or had impacts from inundation/erosion or other threats).

It would be useful for the reader to know the approximate cost for the materials to build one unit, especially as it is said to be cost effective. 

AUTHOR RESPONSE: All the plans for creating the circuitry and assembling the units are on the Nerds without Borders website (http://nerdswithoutborders.net/index.php?title=Turtle_Sense/Phase_Two). The cost of parts is approximately $300 per unit to build independently ($50/sensor and $250/communication tower). Each nest needs its own sensor for the duration of the incubation (placed on the day the nest is laid), though the communication tower does not need to be installed until hatching is more imminent, so can be moved from nest to nest as needed). This information is now in the methods section.

In addition, does the unit itself (that is placed into the nest) give off any heat / have potential to impact hatch success?

AUTHOR RESPONSE: The sensor that is placed into the nest does not give off any heat.

 It would be useful to compare hatches of nests with TurtleSense to those without (removing nests that were relocated or had impacts from inundation/erosion or other threats).

AUTHOR RESPONSE: During the first several seasons of TurtleSense use, we were in a proof-of-concept mode where we tried and modified many variables. During this time, we placed sensors in both relocated and in situ nests to anecdotally observe any impact. No differences were observed in the number of hatchlings that emerged or overall nest motion tracings, so we did not distinguish between relocated and in situ nests in subsequent seasons when we collected data for this paper.

Line 185 - 186: The sensor sends data to the commications unit when it is ready to upload a report to the internet - It isn't clear what this is based on, is it when the storage reaches a certain limit or is it at regular time intervals?

AUTHOR RESPONSE: The sensor sends data when there are 240 records, which is either one day (using 6 minute records) or 4 hours (using 1 minute records). The methods section has now been updated under “TurtleSense software” to reflect this.

Line 290: "motion readings from 79 nests between 2014-2017.." but on line 236 it states that 74 nests were monitored with TurtleSense between 2013-2017. Are these different nests? please clarify.

AUTHOR RESPONSE: These are the same nests. The lines have been updated to clarify this.

Line 291 - 292: Please add in sample sizes for how many nests were relocated and how many were in-situ. There is also no mention of whether the differences in activities were tested between relocated nests and in-situ nests which is important. Methods on how nest relocations were carried out should be included (was the depth and shape of the natural chamber taken into account etc).

AUTHOR RESPONSE: There were 9 Caretta caretta relocated nests in the data that were used for analysis and the rest were in-situ. The relocated nests were relocated due to unsafe original nest placement near the water’s edge and reburied near the dunes at the same nest depth and with the same egg placement. This section of the methods has now been updated to include this information, as well as information on how the nest relocations were handled. In addition, a disclaimer has been added to the discussion acknowledging that there could be hatchling behavioral differences between relocated and in situ nests, though we did not observe noticeable differences in the motion data.

Line 291 - 297: It is mentioned that data from 79 nests are analysed. 72 are loggerhead and 2 are green, what about the other 5? Is it the olive ridley nests that are mentioned below? clarify.

AUTHOR RESPONSE: These are the same nests. The lines have been updated to clarify this.

Line 296: More detail needed here on what data were analysed, was it just motion or did this include temperature/other measures?

AUTHOR RESPONSE: This paragraph says that motion data was analyzed. TurtleSense does collect temperature, but we did not need to analyze it in order to predict emergence accurately. As such, we did not analyze temperature fluctuations as a predictor of emergence, though we did collect the data and it is available for analysis if another research group is interested in temperature. The manuscript has been updated to explain that in the discussion. 

Results

General comments: It is not always immediately clear when talking about emergence if the authors mean emergence from eggs or emergence from the sand. Please make this clearer throughout, including in Figure 1. Was every infertile nest detected before excavating? How many times was the model correct for predicting the emergence date (from the sand)?

AUTHOR RESPONSE: Thank you for pointing out the confusion in the use of the terms “hatching” and “emergence.” We have updated the use of these terms throughout the manuscript to make them more clear (hatching is always coming out of the eggs within the nest, while emergence is always coming out of the nest onto the sand). 

Every infertile nest was detected before excavating based on the absence of hatching activity. The predictive model was generated based on the described data, and can be used whenever hatching activity is present. The ability of the predicted emergence date to continually be narrowed based on hatching data is the strength of this model, as the 3.8 day window can be shortened to a 1 or 2 day window (based on nest depth) when the real-time data shows that hatching activity has ceased within the nest. This information has been added to the discussion.

Fig.1: Needs to be higher quality with larger font used.

AUTHOR RESPONSE: We have revised this figure to be higher quality with larger font.

Lines 373 - 374: In methods it should be added that nests were checked (and how frequently) and that depressions were noted.

AUTHOR RESPONSE: Nests were checked daily each morning and depressions were noted when present. This information has been added to the methods. Emergence count methods were also added.

Line 376: Repetitive...reword.

AUTHOR RESPONSE: This has been reworded to be less repetitive.

Line 379 - 380: This is the first time introducing 'popping' and 'boiling'. Introduce these terms in the introduction if they are to be used as not all readers will understand.

AUTHOR RESPONSE: To be more clear, we have replaced “popping” with “hatching” throughout the manuscript and defined the terms “boil” and “trickle” when they first appear in the manuscript. 

Fig. 2: Day 47 seems too early to be excavating eggs, given hatching was ccurring between 47.5-50.7 days. What was the reason to excavate at this point?

AUTHOR RESPONSE: In this Figure, Day 47 (B) and Day 46 (C) are both too early to excavate based on standard protocols, but this figure illustrates how the TurtleSense system was able to assist when standard methods are challenged (due to the impending overwash from the tropical storm). Both nests were in vulnerable areas and so decisions were made to excavate them early. The figure legend has been updated to include this information.

Line 402: Explain what 'swings' are.

AUTHOR RESPONSE: We have updated the beginning of the results section to more clearly describe what the data looked like in each stage, including defining “swings.” Early on during the first in-the-nest trials of the Turtle Sense technology, the developers had no idea what the data would look like. The first nests started recording data early on, starting at day one or day two. For many weeks the data showed slow oscillations of the readings at a very low level. Because we did not know what these oscillations were, we tried to correlate them (unsuccessfully) with tides, temperature and weather conditions. We wondered if the oscillations might be related to the developing eggs. To test that theory, a sensor was buried in the sand at the same depth of a typical nest, in a location where we knew that there was no nest. The readings we measured were virtually identical to the sensors in a nearby nest, so it was assumed that the readings were the result of background noise. The source of the noise might be a combination of temperature, surf noise, beach traffic, blowing sand, and weather. The actual source was determined to be unimportant from that point on. What became important were readings that did not fit the gentle pattern of those oscillations. 

There were several new data patterns that emerged over time. One pattern was a sharp sudden discontinuity in the readings of the background noise. Over time we were able to correlate these with nest overwashes. The sudden addition of water to the sand around the sensor may have changed the acoustic properties of the sand causing the discontinuity.

Another new pattern was data that showed very large spikes in readings – many times higher than the background noise – for a short amount of time and occurring at random times during gestation. These were frequently correlated with predation events. These predatory events by animals such as ghost crabs could be confirmed in most cases by visual examination of the nests the subsequent day. We also noticed similar readings after nest checks by members of the National Park’s Turtle Patrol. 

With the very first nest we kept looking for data that would indicate nest hatching activity after 40 days of gestation. We quickly identified periods of activity that looked nothing like the other patterns we had seen. These “periods of activity” were identified by data swings that were both higher and lower than the slow background noise oscillations, but not nearly as big in magnitude or as sudden as the predation events. The oscillations were more frequent than the background noise oscillations, occurring at least 3 times a day or more. These oscillations stayed roughly in the same range and frequency for the several days they were noticed. From our first nest we correlated this activity with hatching activity, since it was always followed by hatchlings emerging from the nest a few days later.

The onset of a period of activity was often preceded by a slow drop in the readings. This was not noticed in all of the nests, and we don’t know what this might indicate, if anything.

The conclusion of a period of activity was defined by an abrupt stop in the oscillations. The data readings from that point on remained relatively flat or once again looked like background noise. No patterns were noticed in the data from that point until the hatchlings emerged.

Fig.3: Needs to be clearer on the diagram that the +1 and +2 days are used instead of, not as well as, the original 1.5 day estimation.

AUTHOR RESPONSE: We have updated this figure to make the phases and the predictions more clear.

Line 434: How was nest depth measured? Was it depth to the base of the chamber or the top? Should be in methods.

AUTHOR RESPONSE: Nest depth is measured to the top of the nest chamber. This has now been included in the methods sections.

Discussion

General comments: It should be mentioned that although this is a useful tool for many reasons, other initiatives, such as turtle friendly lighting' should be encouraged (or enforced as many ) so that interference to nests can be reduced in general, rather than allowing the issue to persist and increase. Terminology is switched between 'baby turtles', 'hatchlings' and 'turtles'. Best to choose one - I prefer 'hatchlings' as a more scientific term to baby turtles.

AUTHOR RESPONSE: We agree that the inclusion of turtle friendly lighting initiatives is beneficial and we have updated the intro and the discussion to reflect the importance of this issue. We have changed the terminology throughout the manuscript to read as “hatchlings” in most cases for consistency.

Line 472: I somewhat disagree that this is reliable estimate of when hatchlings are emerging onto the beach as it could be up to 7.5 days. It is providing more of a reliable estimate than previous methods but is still not a method that could be relied upon really for an exact emergence.

AUTHOR RESPONSE: While we agree that our data still cannot allow us to predict a trickle vs a boil, the ability of the predicted emergence date to continually be narrowed based on hatching data is the strength of this model, as the 3.6 day window can be shortened emergence the following day (+1) or in 2 days (+2) day window (based on nest depth) when the real-time data shows that hatching activity has ceased within the nest. This allows protection barriers to be erected in real time instead of solely based on the day count from the time the nest is laid, shortening the amount of time that the nest needs to be closely monitored. This information has been added to the first paragraph in the discussion for clarity.

Line 525: It needs to be noted that there are likely differences in relocated and in-situ nests as relocated nests will not entirely reflect natural conditions.

AUTHOR RESPONSE: Yes, it is possible that differences exist in hatching and emergence behaviors between in situ and relocated nests, but we did not observe noticeable differences in our activity monitoring. This information has been added to the discussion section.

Line 576: This is incorrect based on the report referenced. The report states that the average hatch success ranged from 33 - 77% (based on the month) with lower success being linked to storm erosion and nest inundation (which is different to nest infertility). The report referenced does not mention nest infertility.

AUTHOR RESPONSE: Thank you for your review of this reference. The data presented in the manuscript was updated to better reflect the descriptive statistics in the reference.

---

## [Decision Letter · Decision Letter 1]

21 Jun 2022

PONE-D-21-39727R1The secret life of baby turtles: A novel system to predict hatchling emergence, detect infertile nests, and remotely monitor sea turtle nest eventsPLOS ONE

Dear Dr. Clabough,

Thank you for submitting your manuscript to PLOS ONE. Your revised manuscript has been evaluated by the Academic Editor and one of the previous subject experts that reviewed the original version. We both feel that the manuscript has been much improved, and that you should be commended for your attention to detail in the response. However, there are a few editorial changes that should be considered prior to publication. Therefore, we invite you to submit a revised version of the manuscript that addresses the minor points raised by the subject expert in the review below. Please submit your revised manuscript by Aug 05 2022 11:59PM. If you will need more time than this to complete your revisions, please reply to this message or contact the journal office at plosone@plos.org. Please include the following items when submitting your revised manuscript:A rebuttal letter that responds to each point raised by the academic editor and reviewer(s). You should upload this letter as a separate file labeled 'Response to Reviewers'.A marked-up copy of your manuscript that highlights changes made to the original version. You should upload this as a separate file labeled 'Revised Manuscript with Track Changes'.An unmarked version of your revised paper without tracked changes. You should upload this as a separate file labeled 'Manuscript'.If applicable, we recommend that you deposit your laboratory protocols in protocols.io to enhance the reproducibility of your results. Protocols.io assigns your protocol its own identifier (DOI) so that it can be cited independently in the future. For instructions see: https://journals.plos.org/plosone/s/submission-guidelines#loc-laboratory-protocols. Additionally, PLOS ONE offers an option for publishing peer-reviewed Lab Protocol articles, which describe protocols hosted on protocols.io. Read more information on sharing protocols at https://plos.org/protocols?utm_medium=editorial-email&utm_source=authorletters&utm_campaign=protocols.

We look forward to receiving your revised manuscript.

Kind regards,

Christopher M. Somers

Academic Editor

PLOS ONE

Journal Requirements:

Reviewers' comments:

Reviewer's Responses to Questions

**Comments to the Author**

1. If the authors have adequately addressed your comments raised in a previous round of review and you feel that this manuscript is now acceptable for publication, you may indicate that here to bypass the “Comments to the Author” section, enter your conflict of interest statement in the “Confidential to Editor” section, and submit your "Accept" recommendation.

Reviewer #2: (No Response)

2. Is the manuscript technically sound, and do the data support the conclusions?

Reviewer #2: Yes

3. Has the statistical analysis been performed appropriately and rigorously? 

Reviewer #2: Yes

4. Have the authors made all data underlying the findings in their manuscript fully available?

Reviewer #2: Yes

5. Is the manuscript presented in an intelligible fashion and written in standard English?

Reviewer #2: Yes

6. Review Comments to the Author

Reviewer #2: Thank you to the authors for making the changes and responding to all the comments. The manuscript has been much improved by the changes made, however, it still requires further minor revisions. I have quoted line numbers from the clean version of the revised manuscript:

Line 56) “almost” to the day

Lines 58 - 59) It is more appropriate to say “Our results suggest that motion plays a large role in…. instead of saying you hypothesise as it is from your findings that bring you to this.

Same on Line 650-652. Suggested change “As motion ceases within the nest, it is possible that this final quieting down might be the cue to hatchlings that all their siblings have hatched and it is time to leave the nest”

Lines 70 – 71) Reword and expand, something like: “ Sea turtle populations worldwide are in decline because of an assortment of threats, most of which are attributed to anthropogenic pressure including development and loss of nesting habitat, fishing by-catch, poaching, climate change, ocean pollution”.

Line 75) If you make the above change then include “also highly” to this sentence: “Newly hatched sea turtles (hatchlings) are also highly susceptible to…..”

Line 72) “Conservation efforts largely concentrate on protecting juveniles…..” Move this to start of 3rd paragraph and further expand “….because of the survival rates of hatchlings are low (reference)”

Lines 339 – 342) Rephase suggested: “Using TurtleSense, motion readings from 79 nests were collected over a period of 5 years….(include numbers as you have done)….From these, the 72 C caretta nests from Hatteras Island were analysed for this study (2014 n=20)…etc.”

Lines 345-346) Rephase suggested: “Graphs were automatically generated with sensor data once a day for the first 40 days and then every four hours after that so that nest data could be analysed for patterns of activity throughout development.”

Line 354) include section about relocated number of nests on line 342 after you give the total nest counts instead of here.

Line 417) Include % of nests that it did appear in

Line 523) I still don’t see where the nest depth description is in the methods. Please include in methods how the measure was taken (to top of clutch).

This sentence should be included in methods not results, and change “is” to “was”:

“Nest depth was recorded when the sensor was placed”

Then under the results section for nest depth, the paragraph can start like: “The mean nest depth was…..”.

Line 571) Sub headings here would be useful for each phase, for example:

Incubation: The first period of initial incubation appears to show little motion besides that of background noise.

Prehatch:

Hatching:

Posthatch:

Line 651) “cue” not clue

7. PLOS authors have the option to publish the peer review history of their article (what does this mean?). If published, this will include your full peer review and any attached files.

Reviewer #2: No

---

## [Author Response · Author response to Decision Letter 1]

12 Aug 2022

August 5, 2022

Dear PLoS ONE,

Thank you to the reviewer for your careful reading of the manuscript. We find your help extremely useful and are grateful for your time. We have made all requested changes, as listed below. 

Thank you,

Erin Clabough

Line 56) “almost” to the day

This change has been made.

Lines 58 - 59) It is more appropriate to say “Our results suggest that motion plays a large role in…. instead of saying you hypothesise as it is from your findings that bring you to this.

This change has been made.

Same on Line 650-652. Suggested change “As motion ceases within the nest, it is possible that this final quieting down might be the cue to hatchlings that all their siblings have hatched and it is time to leave the nest”

This change has been made.

Lines 70 – 71) Reword and expand, something like: “ Sea turtle populations worldwide are in decline because of an assortment of threats, most of which are attributed to anthropogenic pressure including development and loss of nesting habitat, fishing by-catch, poaching, climate change, ocean pollution”.

This change has been made.

Line 75) If you make the above change then include “also highly” to this sentence: “Newly hatched sea turtles (hatchlings) are also highly susceptible to…..”

This change has been made.

Line 72) “Conservation efforts largely concentrate on protecting juveniles…..” Move this to start of 3rd paragraph and further expand “….because of the survival rates of hatchlings are low (reference)”

This paragraph has been reworked and the change has been made.

Lines 339 – 342) Rephase suggested: “Using TurtleSense, motion readings from 79 nests were collected over a period of 5 years….(include numbers as you have done)….From these, the 72 C caretta nests from Hatteras Island were analysed for this study (2014 n=20)…etc.”

This change has been made.

Lines 345-346) Rephase suggested: “Graphs were automatically generated with sensor data once a day for the first 40 days and then every four hours after that so that nest data could be analysed for patterns of activity throughout development.”

This change has been made.

Line 354) include section about relocated number of nests on line 342 after you give the total nest counts instead of here.

This section has been moved.

Line 417) Include % of nests that it did appear in

This section has been updated to say 1/3 of nests.

Line 523) I still don’t see where the nest depth description is in the methods. Please include in methods how the measure was taken (to top of clutch).

This information has been added to the methods (Line 343).

This sentence should be included in methods not results, and change “is” to “was”:

This change has been made.

“Nest depth was recorded when the sensor was placed”

This change has been made.

Then under the results section for nest depth, the paragraph can start like: “The mean nest depth was…..”.

This change has been made (Line 523).

Line 571) Sub headings here would be useful for each phase, for example:

Incubation: The first period of initial incubation appears to show little motion besides that of background noise.

Prehatch:

Hatching:

Posthatch:

These subheadings have been added.

Line 651) “cue” not clue

This change has been made.

---

## [Editor Report · Decision Letter 2]

29 Aug 2022

PONE-D-21-39727R2The secret life of baby turtles: A novel system to predict hatchling emergence, detect infertile nests, and remotely monitor sea turtle nest eventsPLOS ONE

Dear Dr. Clabough,

Thank you for submitting your second revision to PLOS ONE. The manuscript is much improved and reads well. However, there are still several minor changes required. Therefore, we invite you to submit a revised version of the manuscript that addresses the points raised during the review process.

Required Changes:1. Line 69, 120, etc. the authors refer to sea turtles as if they are one species in several places. This needs to be corrected.2. Line 72: this is not a complete sentence; please revise.3. Line 135: informal language used here describing "cork in bottle" situation; please revise.4. Throughout document: "data" is a plural word and needs to be treated that way grammatically at each use in the manuscript. For example, the authors often use "data was" or "data is", when the correct version is "data were" or "data are".5. Throughout the manuscript: the authors often place the punctuation at the end of sentences before the brackets containing the references. This needs to be corrected in every instance.

We look forward to receiving your revised manuscript.

Kind regards,

Christopher M. Somers

Academic Editor

PLOS ONE
---

## [Author Response · Author response to Decision Letter 2]

8 Sep 2022

Dear PLoS ONE,

Thank you to the reviewer for your careful reading of the manuscript. We find your help extremely useful and are grateful for your time. We have made all requested changes, as listed below. 

Thank you,

Erin Clabough

Required Changes:

1. Line 69, 120, etc. the authors refer to sea turtles as if they are one species in several places. This needs to be corrected.

This change has been made, more specifically referring to individual species in current line 69, 118, 119, 124.

2. Line 72: this is not a complete sentence; please revise.

This change has been made to include the word “pollution.”

3. Line 135: informal language used here describing "cork in bottle" situation; please revise.

This change has been made to remove the informal language in line 139.

4. Throughout document: "data" is a plural word and needs to be treated that way grammatically at each use in the manuscript. For example, the authors often use "data was" or "data is", when the correct version is "data were" or "data are".

This change has been made in numerous locations throughout the document.

5. Throughout the manuscript: the authors often place the punctuation at the end of sentences before the brackets containing the references. This needs to be corrected in every instance.

This change has been made throughout the document.

---

## [Editor Report · Decision Letter 3]

12 Sep 2022

The secret life of baby turtles: A novel system to predict hatchling emergence, detect infertile nests, and remotely monitor sea turtle nest events

PONE-D-21-39727R3

Dear Dr. Clabough,

We’re pleased to inform you that your manuscript has been judged scientifically suitable for publication and will be formally accepted for publication once it meets all outstanding technical requirements.

Kind regards,

Christopher M. Somers

Academic Editor

PLOS ONE
---

## [Editor Report · Acceptance letter]

4 Oct 2022

PONE-D-21-39727R3 

The secret life of baby turtles:
A novel system to predict hatchling emergence, detect infertile nests, and remotely monitor sea turtle nest events 

Dear Dr. Clabough:

I'm pleased to inform you that your manuscript has been deemed suitable for publication in PLOS ONE. Congratulations! Your manuscript is now with our production department. 

Kind regards, 

on behalf of

Dr. Christopher M. Somers 

Academic Editor

PLOS ONE